# mRNA stem-loops can pause the ribosome by hindering A-site tRNA binding

Chen Bao[1†], Sarah Loerch[2†], Clarence Ling[1], Andrei A Korostelev[3,4], Nikolaus Grigorieff[2,4*], Dmitri N Ermolenko[1*]

[1]Department of Biochemistry and Biophysics at School of Medicine and Dentistry and Center for RNA Biology, University of Rochester, Rochester, United States; [2]Janelia Research Campus, Howard Hughes Medical Institute, Ashburn, United States; [3]Department of Biochemistry and Molecular Pharmacology, University of Massachusetts Medical School, Worcester, United States; [4]RNA Therapeutics Institute, University of Massachusetts Medical School, Worcester, United States

**Abstract** Although the elongating ribosome is an efficient helicase, certain mRNA stem-loop structures are known to impede ribosome movement along mRNA and stimulate programmed ribosome frameshifting via mechanisms that are not well understood. Using biochemical and single-molecule Förster resonance energy transfer (smFRET) experiments, we studied how frameshift-inducing stem-loops from *E. coli dnaX* mRNA and the *gag-pol* transcript of Human Immunodeficiency Virus (HIV) perturb translation elongation. We find that upon encountering the ribosome, the stem-loops strongly inhibit A-site tRNA binding and ribosome intersubunit rotation that accompanies translation elongation. Electron cryo-microscopy (cryo-EM) reveals that the HIV stem-loop docks into the A site of the ribosome. Our results suggest that mRNA stem-loops can transiently escape the ribosome helicase by binding to the A site. Thus, the stem-loops can modulate gene expression by sterically hindering tRNA binding and inhibiting translation elongation.

*For correspondence:
niko@grigorieff.org (NG);
Dmitri_Ermolenko@urmc.
rochester.edu (DNE)

†These authors contributed equally to this work

## Introduction

During translation elongation, the ribosome moves along mRNA in a codon-by-codon manner while the mRNA is threaded through the mRNA channel of the small ribosomal subunit. In the bacterial ribosome, the mRNA channel accommodates 11 nucleotides downstream of the first (+1) nucleotide of the P-site codon (*Takyar et al., 2005*; *Yusupova et al., 2001*). The translating ribosome must unfold mRNA secondary structure to feed single-stranded mRNA through the narrow mRNA channel. The ribosome was shown to be a processive helicase, which unwinds three basepairs per translocation step (*Qu et al., 2011*; *Takyar et al., 2005*; *Wen et al., 2008*). Accordingly, transcriptome-wide ribosome profiling analysis demonstrated that most of the secondary structure elements within coding regions of mRNAs do not influence the rate of translation elongation (*Del Campo et al., 2015*).

Although the elongating ribosome is an efficient helicase, certain mRNA stem-loop structures are known to pause or stall ribosome movement along mRNA. mRNA stem-loop structures can induce ribosome stalling that results in accumulation of truncated polypeptides (*Yan et al., 2015*) and no-go mRNA decay (*Doma and Parker, 2006*). In addition, evolutionarily conserved mRNA stem-loops trigger programmed translation pauses. For example, the α subunit of the signal recognition particle receptor is co-translationally targeted to the endoplasmic reticulum membrane by a mechanism that requires a translational pause induced by an mRNA stem-loop structure (*Young and Andrews,*

*1996*). Ribosome pausing induced by mRNA hairpins and pseudoknots accompanies −1 programmed ribosome frameshifting (PRF), which controls expression of a number of proteins in bacteria, viruses and eukaryotes (*Caliskan et al., 2015*). In particular, −1 PRF regulates synthesis of DNA polymerase III in bacteria (*Tsuchihashi and Kornberg, 1990*); HIV cytokine receptor CCR5 in higher eukaryotes (*Belew et al., 2014*); gag-pol proteins in retroviruses, including Human Immunodeficiency Virus (HIV) (*Jacks et al., 1988*); and C-terminally extended polyprotein in coronaviruses, including SARS-CoV-2, which caused the COVID-19 pandemic (*Dinman, 2012*; *Kelly and Dinman, 2020*).

−1 PRF requires the presence of two signals in an mRNA: the heptanucleotide slippery sequence XXXYYYZ (where X and Z can be any nucleotide and Y is either A or U) and a downstream frameshift stimulating sequence (FSS). The FSS is an RNA hairpin or a pseudoknot (*Atkins et al., 2001*). The slippery sequence allows cognate pairing of the P-site and A-site tRNAs in both 0 and −1 frames and thus makes frameshifting thermodynamically favorable (*Bock et al., 2019*). The mechanism by which FSS stimulates frameshifting is less clear. A number of studies have shown that FSSs inhibit the rate of translocation of the A- and P-site tRNAs basepaired with the slippery sequence by at least one order of magnitude (*Caliskan et al., 2014*; *Caliskan et al., 2017*; *Chen et al., 2014*; *Kim et al., 2014*) and thus produce ribosome pauses (*Caliskan et al., 2014*; *Caliskan et al., 2017*; *Chen et al., 2014*; *Kim et al., 2014*; *Kontos et al., 2001*; *Lopinski et al., 2000*; *Somogyi et al., 1993*; *Tu et al., 1992*).

It remains puzzling why certain stem-loops including FSSs induce ribosome pausing in spite of the ribosome helicase activity. Slow unwinding of secondary structure, to which ribosome pausing is often attributed, is unlikely to account for the extent of translation inhibition induced by FSSs. Single-molecule experiments showed that translocation through three GC basepairs is only 2 to 3-fold slower than translocation along a single-stranded codon (*Chen et al., 2013*; *Desai et al., 2019*; *Qu et al., 2011*), indicating that the stability of the three basepairs adjacent to the mRNA channel has a relatively moderate effect on translocation rate. Consistent with this idea, neither the thermodynamic stability of the entire FSS nor the stability of the basepairs adjacent to the mRNA entry channel fully correlate with the efficiency of frameshifting (*Chen et al., 2009*; *Hansen et al., 2007*; *Mouzakis et al., 2013*; *Ritchie et al., 2012*). Hence, FSSs and other stem-loops inducing ribosome pausing may perturb translation elongation by distinct mechanisms that are not well understood.

Here we use a combination of smFRET, biochemical assays and cryo-EM to investigate how FSSs from *E. coli dnaX* mRNA and the *gag-pol* transcript of HIV pause the ribosome. We asked whether these bacterial and viral mRNA sequences, which form stem-loops of similar lengths, act via a similar mechanism. In agreement with previous studies (*Caliskan et al., 2014*; *Caliskan et al., 2017*; *Chen et al., 2014*; *Choi et al., 2020*; *Kim et al., 2014*), we detected FSS-induced inhibition of tRNA/mRNA translocation. We also observed that FSSs inhibit A-site tRNA binding. Cryo-EM analysis of the 70S ribosome bound with FSS-containing mRNA revealed that the FSS docks into the A site of the ribosome and sterically hinders tRNA binding. Occlusion of the ribosomal A site by an mRNA stem-loop may be a common strategy by which mRNA stem-loops induce ribosome pausing to modulate gene expression.

## Results

### dnaX FSS inhibits intersubunit rotation during translation along the slippery sequence

We investigated how the interaction of FSSs with the ribosome affects cyclic forward and reverse rotations between ribosomal subunits that accompany each translation elongation cycle (*Frank and Gonzalez, 2010*). Following aminoacyl-tRNA binding to the ribosomal A site and peptide-bond formation, the pre-translocation ribosome predominantly adopts a rotated (R) conformation (*Aitken and Puglisi, 2010*; *Cornish et al., 2008*; *Ermolenko et al., 2007*). In this conformation, the small ribosomal subunit (the 30S subunit in bacteria) is rotated by 7–9° relative to the large subunit (the 50S subunit) (*Dunkle et al., 2011*; *Frank and Agrawal, 2000*; *Frank and Gonzalez, 2010*), and two tRNAs adopt the intermediate hybrid states (*Blanchard et al., 2004b*; *Moazed and Noller, 1989*; *Valle et al., 2003*). EF-G-catalyzed mRNA/tRNA translocation on the small subunit is coupled to the reverse rotation of the ribosomal subunits relative to each other, restoring the nonrotated

(NR) conformation in the post-translocation ribosome (*Aitken and Puglisi, 2010*; *Ermolenko et al., 2007*; *Ermolenko and Noller, 2011*).

To probe the effect of FSSs on intersubunit rotation accompanying translation, we employed a model dnaX_Slip mRNA that was derived from the *E. coli dnaX* transcript (*Figure 1*). *dnaX* mRNA encodes the τ and γ subunits of DNA polymerase III. The γ subunit is produced by a −1 PRF event that occurs with 50–80% efficiency. We chose *dnaX* mRNA because it is one of the most extensively studied −1 PRF systems that has been investigated using both ensemble and single-molecule kinetic approaches (*Caliskan et al., 2017*; *Chen et al., 2014*; *Choi et al., 2020*; *Kim et al., 2014*; *Kim and Tinoco, 2017*). The model dnaX_Slip mRNA contained a Shine-Dalgarno (SD, ribosome-binding site) sequence, a short ORF with the slippery sequence AAAAAAG and a downstream 10 basepair-long FSS mRNA hairpin, which together program −1 PRF in *dnaX* mRNA (*Larsen et al., 1997*; *Figure 1*). In addition, upstream of the SD sequence, the dnaX_Slip mRNA contained a 25 nucleotide-long sequence complementary to a biotin-derivatized DNA oligonucleotide used to tether the mRNA to a microscope slide for smFRET experiments (*Figure 1*). The beginning of the ORF encodes Met-Val-Lys-Lys-Arg in 0 frame and Met-Val-Lys-Lys-Glu in −1 frame.

We determined the efficiency of −1 PRF on the dnaX_Slip mRNA during translation along the slippery sequence via the filter-binding assay. To that end, ribosomes bound with dnaX_Slip mRNA and P-site *N*-Ac-Val-tRNA$^{Val}$ were incubated with EF-G•GTP, EF-Tu•GTP, Lys-tRNA$^{Lys}$, Arg-tRNA$^{Arg}$ (binds in 0 frame) and [$^3$H]Glu-tRNA$^{Glu}$ (binds in −1 frame). Consistent with previous publications (*Caliskan et al., 2017*; *Kim and Tinoco, 2017*; *Larsen et al., 1997*), we observed a frameshifting efficiency of ~60% (*Figure 1—figure supplement 1*). When ribosomes were programmed with the

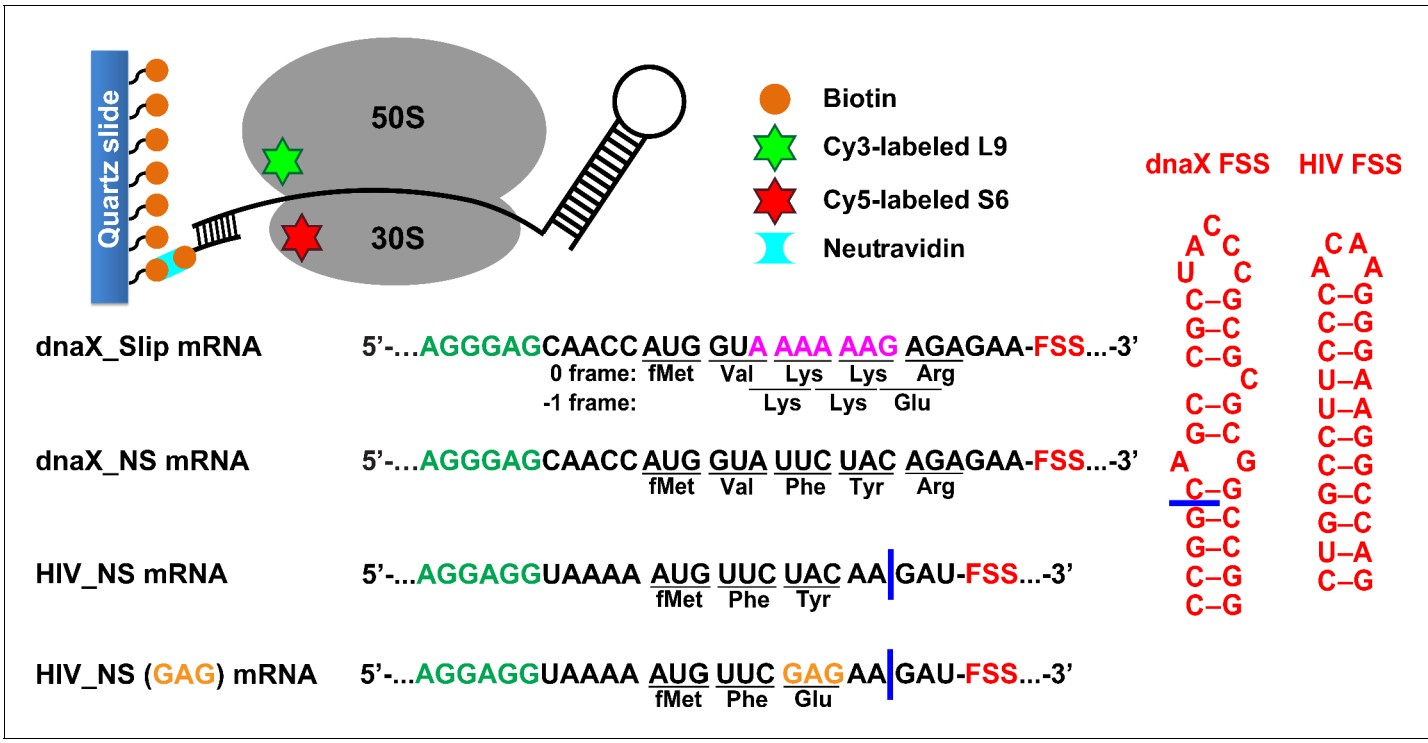

**Figure 1.** Experimental design. The effect of frameshift-inducing mRNA stem-loops on translation elongation was studied using FRET between cy5 (red) and cy3 (green) attached to 30S protein S6 and 50S protein L9, respectively. S6-cy5/L9-cy3 ribosomes were immobilized on quartz slides using neutravidin and biotinylated DNA oligomers annealed to the mRNA. dnaX_Slip mRNA contains an internal SD sequence (green), a slippery sequence (magenta) and an FSS (red). In the non-slippery (NS) dnaX and HIV mRNAs, the slippery sequences were replaced by non-slippery codons. Two different HIV_NS mRNAs contain either a UAC or a GAG (orange) codon. Corresponding polypeptide sequences are shown below each mRNA. The ΔFSS mRNAs are truncated as indicated by blue bars.

The online version of this article includes the following figure supplement(s) for figure 1:

**Figure supplement 1.** FSS and slippery sequence of *dnaX* mRNA stimulate −1 RPF.

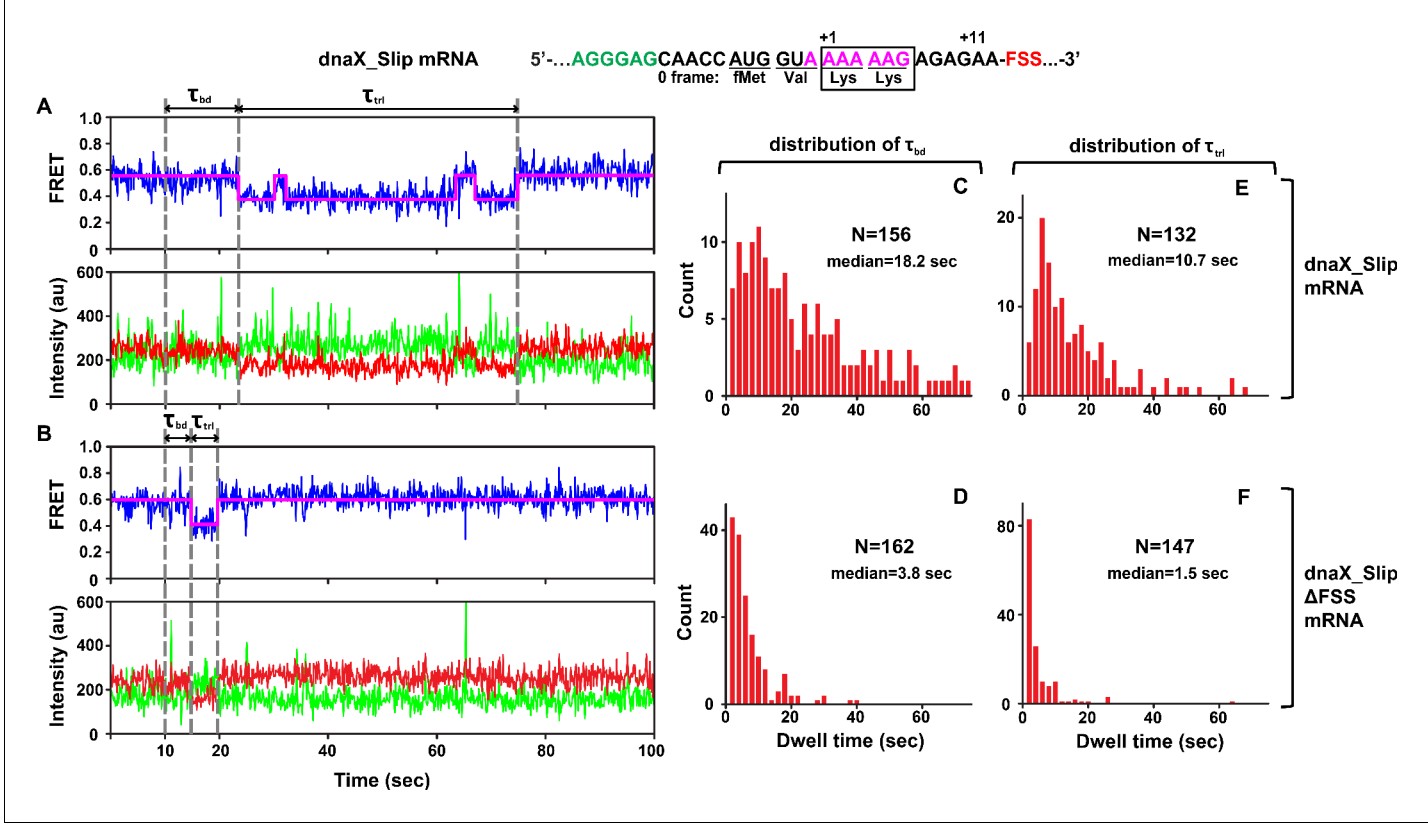

**Figure 2.** DnaX FSS slows ribosome intersubunit rotation. S6-cy5/L9-cy3 ribosomes containing P-site *N*-Ac-Val-Lys-tRNA[Lys] were programmed with either dnaX_Slip (A, C, E) or dnaX_Slip ΔFSS (B, D, F) mRNAs. After 10 s of imaging, EF-Tu•GTP•Lys-tRNA[Lys] and EF-G•GTP were co-injected into the flow-through chamber. (A–B) Representative smFRET traces show cy3 fluorescence (green), cy5 fluorescence (red), FRET efficiency (blue) and the HHM fit of FRET efficiency (magenta). $\tau_{bd}$ is the dwell time between the injection and Lys-tRNA[Lys] binding to the A site, which corresponds to the transition from NR (0.6 FRET) to R (0.4 FRET) state of the ribosome. $\tau_{trl}$ is the dwell time between A-site binding of Lys-tRNA[Lys] and EF-G-catalyzed tRNA translocation, which corresponds to the transition from R to the stable (i.e. lasting over 4 s) NR state of the ribosome. The full-length views of smFRET traces are shown in *Figure 2—figure supplement 1A,B*. (C–F) Histograms (2 s binning size) compiled from over 100 traces show the distributions and median values of $\tau_{bd}$ and $\tau_{trl}$. N indicates the number of FRET traces assembled into each histogram.

The online version of this article includes the following figure supplement(s) for figure 2:

**Figure supplement 1.** Both A-site tRNA binding and translocation are hindered by the dnaX FSS positioned at the entrance of mRNA channel.

truncated dnaX_Slip ΔFSS mRNA, which lacks the FSS (*Figure 1*), the efficiency of −1 PRF decreased to ~25%, demonstrating that the FSS stimulates ribosome frameshifting.

To follow intersubunit rotation during translation along the slippery sequence of the dnaX_Slip mRNA, we measured smFRET between fluorophores attached to the 50S protein L9 and the 30S protein S6. The NR and R conformations of the ribosome have been shown to correspond to 0.6 and 0.4 FRET states of S6-cy5/L9-cy3 FRET pair (*Cornish et al., 2008*; *Ermolenko et al., 2007*).

We asked whether dnaX FSS positioned near the entrance of the mRNA channel perturbs ribosome intersubunit dynamics during frameshifting. To this end, we monitored elongation on S6-cy5/L9-cy3 ribosomes bound with P-site *N*-Ac-Val-Lys-tRNA[Lys] and dnaX_Slip mRNA immobilized on a microscope slide (*Figure 1*). In this ribosome complex, the second Lys codon of the slippery sequence is positioned in the A site, and the FSS is expected to be one nucleotide downstream of the entrance to the mRNA channel (*Yusupova et al., 2001*; *Zhang et al., 2018*). Consistent with previous reports, ribosomes containing P-site peptidyl-tRNA (*N*-Ac-Val-Lys-tRNA[Lys]) are predominately in the NR (0.6 FRET) state (*Figure 2A*, *Figure 2—figure supplement 1A*; *Cornish et al., 2008*; *Ermolenko et al., 2007*). After 10 s of imaging, EF-Tu•GTP•Lys-tRNA[Lys] and EF-G•GTP were injected to bind Lys-tRNA[Lys] to the second Lys codon of the slippery sequence and induce tRNA/mRNA translocation. After the injection, the ribosomes showed an NR (0.6 FRET)-to-R (0.4 FRET) transition

(*Figure 2A*, *Figure 2—figure supplement 1A*). The transpeptidation reaction and subsequent movement of tRNAs into hybrid states are typically much faster than tRNA binding to the A site (*Blanchard et al., 2004a*; *Johansson et al., 2008*; *Juette et al., 2016*; *Rodnina and Wintermeyer, 2001*; *Sharma et al., 2016*). Hence, the dwell time between the injection and the transition from the NR (0.6 FRET) to R (0.4 FRET) state, $\tau_{bd}$, primarily reflects the rate of Lys-tRNA$^{Lys}$ binding to the A site of the ribosome.

The subsequent reverse transition from the R to the stable NR state (0.6 FRET lasting over 4 s) indicated translocation of mRNA and tRNA. In 64% of traces, the transition from R to NR was preceded by one or two short-lived excursions from R to NR, characteristic of pre-translocation ribosomes (*Figure 2A*, *Figure 2—figure supplement 1A*). This observation is consistent with published smFRET experiments demonstrating that under dnaX FSS-induced pausing, two Lys tRNAs undergo multiple unproductive fluctuations between the hybrid and classical states (*Kim et al., 2014*).

To further test whether the transition from R to NR indeed corresponds to tRNA translocation, we imaged pre-translocation ribosomes containing deacylated tRNA$^{Lys}$ in the 30 S P site in the absence of EF-G. In this complex, R and NR interconverted at rates of 0.2 sec$^{-1}$ (R to NR) (*Figure 2—figure supplement 1D*) and 0.6 sec$^{-1}$, (NR to R) (*Figure 2—figure supplement 1E*), respectively. Thus, in the absence of EF-G, 95% of pre-translocation ribosomes spent less than 4 s in the NR state. This analysis further supports the interpretation that the transition from R to the stable NR state (i.e. lasting over 4 s) accompanies translocation of tRNAs and mRNA on the small ribosomal subunit (*Figure 2A*, *Figure 2—figure supplement 1A*). Hence, the dwell time $\tau_{trl}$ between the first NR to R and R to NR transitions in our injection experiments corresponds to the translocation rate.

$\tau_{bd}$ and $\tau_{trl}$ of dnaX_Slip mRNA programmed ribosomes were remarkably long with median values of 18.2 s and 10.7 s, respectively (*Figure 2C,E*), suggesting inefficient tRNA$^{Lys}$ binding and translocation. Moreover, actual median values of $\tau_{bd}$ and $\tau_{trl}$ are likely longer because Cy5 photobleaching occurring at the rate of 0.02 sec$^{-1}$ leaves some ribosomes with long $\tau_{bd}$ and $\tau_{trl}$ undetected. Notably, both $\tau_{bd}$ and $\tau_{trl}$ were broadly distributed (*Figure 2C,E*) and could not be fit to a single exponential decay, suggesting heterogeneity within the ribosome population.

Ribosome complexes assembled with dnaX mRNA lacking the FSS (dnaX_Slip ΔFSS mRNA) showed markedly different behavior in comparison with dnaX_Slip mRNA complexes. When EF-Tu•GTP•Lys-tRNA$^{Lys}$ and EF-G•GTP were added to S6-cy5/L9-cy3 ribosomes programmed with dnaX_Slip ΔFSS mRNA and P-site peptidyl-tRNA (*N*-Ac-Val-Lys-tRNA$^{Lys}$), rapid transition from NR to R was followed by rapid transition to the stable NR state (*Figure 2B*, *Figure 2—figure supplement 1B*). In contrast to dnaX_Slip FSS mRNA programmed ribosomes, which showed spontaneous fluctuations between R and NR states before the transition to the stable post-translocation NR state, only 6% of ribosomes programmed with dnaX_Slip ΔFSS mRNA showed short-lived excursions from R to NR before translocation (*Figure 2—figure supplement 1C*). Median values of $\tau_{bd}$ (3.8 s) and $\tau_{trl}$ (1.5 s) for dnaX_Slip ΔFSS mRNA (*Figure 2D,F*) were 5- and 7-fold shorter, respectively, than those measured in ribosomes programmed with dnaX_Slip mRNA. In agreement with previously published results (*Caliskan et al., 2017*; *Chen et al., 2014*; *Choi et al., 2020*; *Kim et al., 2014*), our data demonstrate that dnaX FSS positioned near the mRNA channel entrance strongly inhibits mRNA/tRNA translocation (*Figure 2E,F*). In addition, our data unexpectedly revealed that dnaX FSS also strongly inhibits A-site tRNA binding during the elongation cycle (*Figure 2C,D*). Because such FSS-induced inhibition of A-site binding has not been observed before, we further explored this phenomenon using smFRET, biochemical and cryo-EM approaches.

## In the presence of non-slippery sequence, the FSS from *dnaX* mRNA stalls the ribosome in the NR conformation

We next asked whether the spacing between the FSS and the mRNA entry channel of the ribosome affects the ability of the FSS to inhibit A-site binding. Because frameshifting changes the position of the FSS relative to the mRNA entry channel of the ribosome, we aimed to decouple FSS-induced ribosome pausing from frameshifting. To that end, we replaced the two consecutive Lys codons of the dnaX slippery sequence with UUC (Phe) and UAC (Tyr) 'non-slippery' codons to create dnaX_NS ('non-slippery') mRNA (*Figure 1*). Mutations in the slippery sequence of *dnaX* were shown to decrease frameshifting efficiency to low (≤5%) or undetectable levels (*Bock et al., 2019*; *Caliskan et al., 2017*; *Kim and Tinoco, 2017*; *Larsen et al., 1997*; *Tsuchihashi and Brown, 1992*).

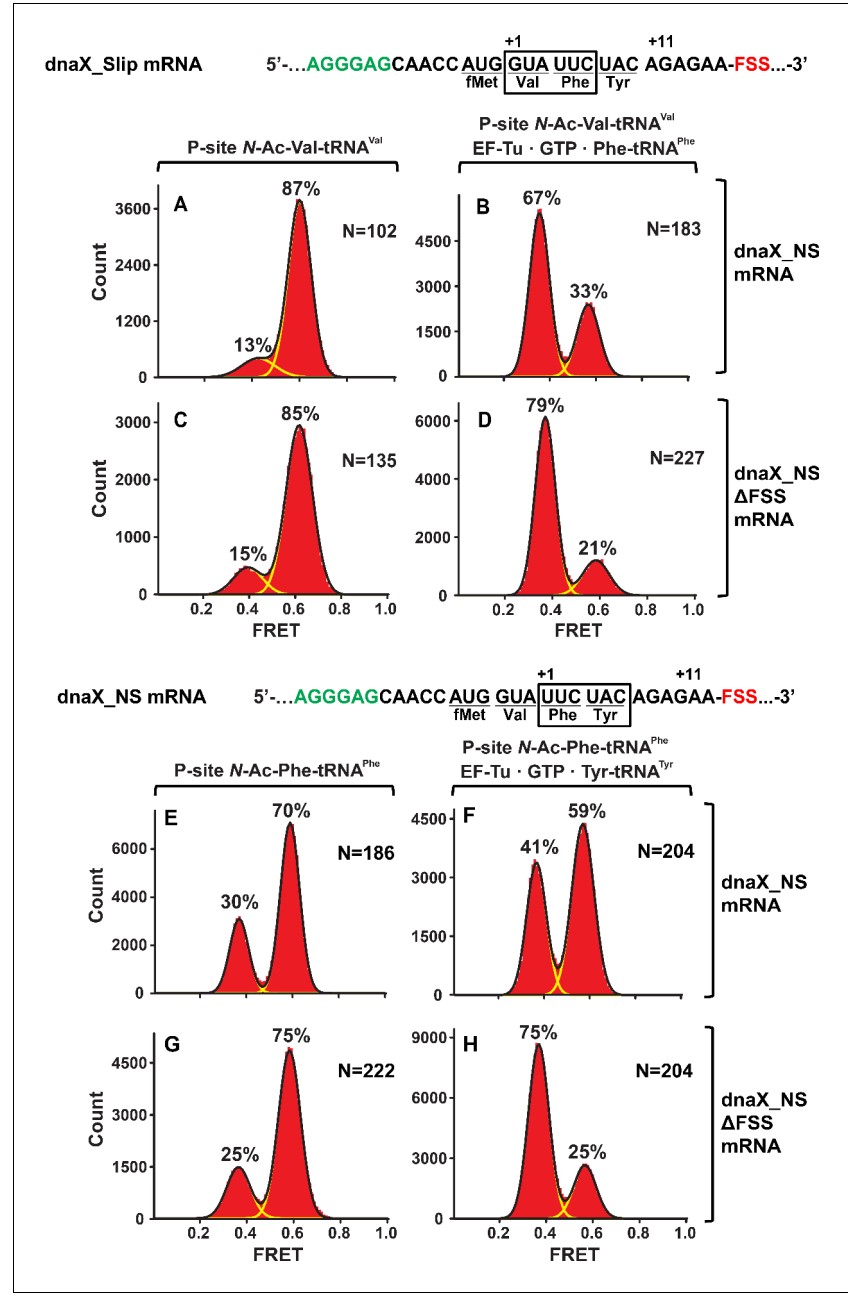

**Figure 3.** In the context of non-slippery codons, the dnaX FSS stalls the ribosome in the NR conformation. Histograms show FRET distributions in S6-cy5/L9-cy3 ribosomes programmed with dnaX_NS (**A–B, E–G**) or dnaX_NS ΔFSS (**C–D, H–J**) mRNA, respectively. Ribosomes were bound with P-site peptidyl tRNA analogs, *N*-Ac-Val-tRNA$^{Val}$ (**A, C**) or *N*-Ac-Phe-tRNA$^{Phe}$ (**E, G**). The ribosomes were then incubated with either EF-Tu•GTP•Phe-tRNA$^{Phe}$ (**B, D**) or EF-Tu•GTP•Tyr-tRNA$^{Tyr}$ (**F, H**) for 5 min and imaged after removal of unbound aminoacyl-tRNAs. Yellow lines show individual Gaussian fits of FRET distributions. Black lines indicate the sum of Gaussian fits. N indicates the number of FRET traces compiled into each histogram. The fractions of the ribosome in R and NR conformations are shown above the corresponding 0.4 and 0.6 Gaussian peaks, respectively.

When EF-Tu•GTP•Phe-tRNA$^{Phe}$ was added to ribosomes with the FSS positioned four nucleotides away from the entry of the mRNA channel, the ribosome population converted from a predominant NR (*Figure 3A,C*) to an R conformation (*Figure 3B,D*) as expected for the 'normal' elongation cycle. By contrast, when EF-Tu•GTP•Tyr-tRNA$^{Tyr}$ was added to ribosomes with the FSS positioned one nucleotide away from the entry of mRNA channel, the majority (60%) of ribosomes remained in the

NR conformation (*Figure 3F*). Hence, the encounter of the ribosome mRNA entry channel with the FSS inhibits conversion of the ribosome from the NR to R conformation, which accompanies tRNA binding. Indeed, when EF-Tu•GTP•Tyr-tRNA$^{Tyr}$ was added to ribosomes programmed with dnaX_NS ΔFSS mRNA, which lacks the FSS, the ribosome population was converted from predominately NR (*Figure 3G*) to R conformation (*Figure 3H*).

## The FSS from HIV also stalls the ribosome in the NR conformation

We considered if other frameshift-inducing mRNA stem-loops can induce ribosome stalling in the NR conformation, similar to the FSS from dnaX mRNA. We chose to study the 12 basepair-long RNA hairpin from HIV (*Figure 1*) that in combination with the slippery sequence UUUUUUA, induces −1 PRF with 5–10% efficiency to produce the Gag-Pol polyprotein. mRNAs containing the slippery sequence and HIV FSS undergo frameshifting in bacterial (*E. coli*) ribosomes in vitro and in vivo at frequencies comparable to those observed for HIV frameshifting in eukaryotic translation systems (*Brunelle et al., 1999*; *Korniy et al., 2019a*; *Léger et al., 2004*; *Mazauric et al., 2009*). The FSS from HIV can be studied in *E. coli*, analogous to −1 PRF on mRNA derived from another eukaryotic virus (avian infectious bronchitis virus, IBV) that could also be reconstituted in the *E. coli* translation system (*Caliskan et al., 2014*), suggesting a common mechanism of frameshifting and ribosomal stalling induced by FSS in bacteria and eukaryotes.

Similar to dnaX_NS mRNA, we designed an HIV_NS mRNA that contained a 25-nucleotide sequence complementary to a biotinylated DNA handle, the SD sequence, and a short ORF containing the FSS. The original HIV sequence UUU UUA GGG including slippery codons was replaced with AUG UUC UAC 'non-slippery' codons to delineate the FSS-induced ribosome pausing from frameshifting (*Figure 1*). When EF-Tu•GTP•Phe-tRNA$^{Phe}$ was incubated with ribosomes spaced three nucleotides away from the HIV FSS, the conformation of the ribosome population shifted from predominantly NR (*Figure 4A,C*) to the R conformation (*Figure 4B,D*). By contrast, when EF-Tu•GTP•Tyr-tRNA$^{Tyr}$ was added to ribosomes with the HIV FSS at the entry channel (*Figure 4E*), the majority (60%) of ribosomes remained in the NR conformation *Figure 4F*. When EF-Tu•GTP•Tyr-tRNA$^{Tyr}$ was added to ribosomes programmed with HIV_NS ΔFSS mRNA, which lacks the FSS, the ribosome population converted from predominately NR (*Figure 4G*) to the R conformation (*Figure 4H*). Therefore, similar to the FSS from *dnaX*, upon encountering the ribosome, the FSS from HIV stalls the ribosome in the NR conformation.

To test whether identities of A-site codon and A-site tRNA affect the observed ribosome stalling in the NR conformation, we made HIV_NS (GAG) mRNA, in which the original UAC (Tyr) codon of HIV_NS mRNA was replaced with a GAG (Glu) codon (*Figure 1*). The resulting complexes behaved similarly to the complexes assembled with the original HIV_NS mRNA (*Figure 5A–D*). The majority of ribosomes (60%) with the FSS at the mRNA entry channel remained in the NR conformation after addition of EF-Tu•GTP•Glu-tRNA$^{Glu}$ (*Figure 5B*) while ribosomes programmed with HIV_NS (GAG) ΔFSS mRNA switched to the R conformation (*Figure 5D*). Thus, we show that the FSS-induced inhibition of tRNA binding is independent of A-site codon identity.

Next, we tested whether the stalling in the NR conformation observed with dnaX_NS and HIV_NS mRNAs was due to mRNA frameshifting that prevented A-site binding of Tyr-tRNA$^{Tyr}$ (or Glu-tRNA-$^{Glu}$ in the case of ribosomes programmed with HIV_NS (GAG) mRNA). S6-cy5/L9-cy3 ribosomes, which were programmed with either dnaX_NS (UAC) or HIV_NS (UAC) mRNA and bound with P-site *N*-Ac-Phe-tRNA$^{Phe}$ (*Figure 5—figure supplement 1A,C*), were incubated for 5 min with EF-Tu•GTP and 150-fold molar excess of total tRNA from *E. coli* aminoacylated with 19 natural amino acids except for Tyr. Incubation with total aa-tRNA (minus Tyr) did not lead to an appreciable increase in the fraction of the R (0.4 FRET) conformation (*Figure 5—figure supplement 1B,D*), indicating the lack of A-site tRNA binding in the absence of Tyr-tRNA$^{Tyr}$. By contrast, as a positive control, just a 30-fold molar excess of total aa-tRNA (minus Tyr) was sufficient to decode an in-frame Glu (GAG) codon in ribosomes programmed with HIV_NS (GAG) ΔFSS mRNA as evident from the conversion of the ribosome population from the NR to R conformation (*Figure 5—figure supplement 1E–G*). Therefore, in the absence of the slippery sequence, FSS-induced frameshifting is negligible and does not account for ribosome stalling in the NR conformation observed in the experiments with ribosomes programmed with dnaX_NS or HIV_NS mRNAs.

HIV and dnaX FSSs placed near the entry to the mRNA channel could stall the ribosome in the NR conformation by either (i) inhibiting A-site tRNA binding, (ii) blocking the peptidyltransfer

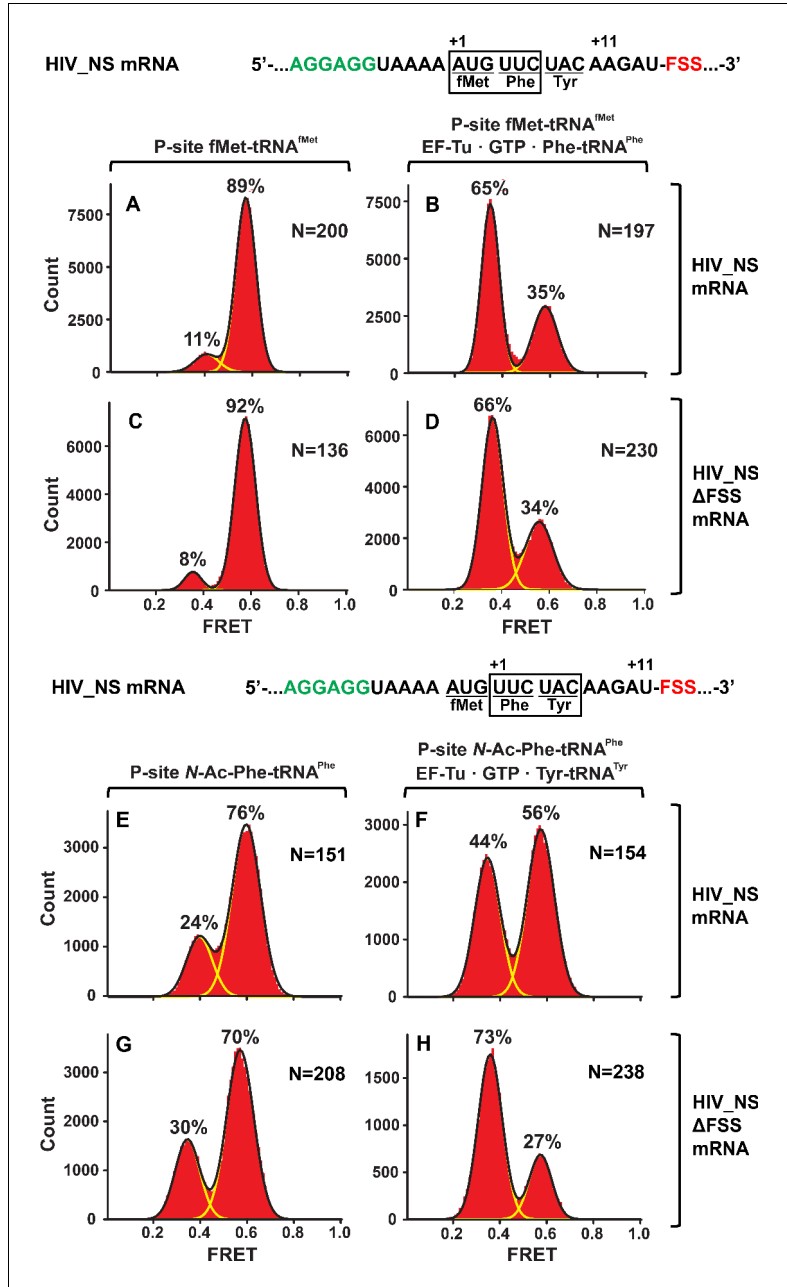

**Figure 4.** In the context of non-slippery codons, the HIV FSS stalls the ribosome in NR conformation. Histograms show FRET distributions in S6-cy5/L9-cy3 ribosomes programmed with HIV_NS (**A–B, E–G**) or HIV_NS ΔFSS (**C–D, H–J**) mRNA, respectively. Ribosomes contained fMet-tRNA$^{fMet}$ (**A, C**) or $N$-Ac-Phe-tRNA$^{Phe}$ (**E, G**) in the P site. The ribosomes were then incubated with either EF-Tu•GTP•Phe-tRNA$^{Phe}$ (**B, D**) or EF-Tu•GTP•Tyr-tRNA$^{Tyr}$ (**F, H**) for 5 min and imaged after removal of unbound aminoacyl-tRNAs. Yellow lines show individual Gaussian fits of FRET distributions. Black lines indicate the sum of Gaussian fits. N indicates the number of FRET traces compiled into each histogram. The fractions of the ribosome in R and NR conformations are shown above the corresponding 0.4 and 0.6 Gaussian peaks, respectively.

reaction after the binding of A-site tRNA or (iii) stabilizing the pre-translocation ribosome in the NR conformation. Pretranslocation-like S6-cy5/L9-cy3 ribosomes containing deacylated P-site tRNA$^{Phe}$ exhibited similar intersubunit dynamics regardless of whether they were programmed with dnaX_NS, dnaX_NS ΔFSS, HIV_NS or HIV_NS ΔFSS mRNAs. These complexes fluctuated between the R (0.4 FRET) and NR (0.6 FRET) states at rates of 0.2–0.3 sec$^{-1}$ (0.4 FRET to 0.6 FRET) and 0.7–0.8 sec$^{-1}$,

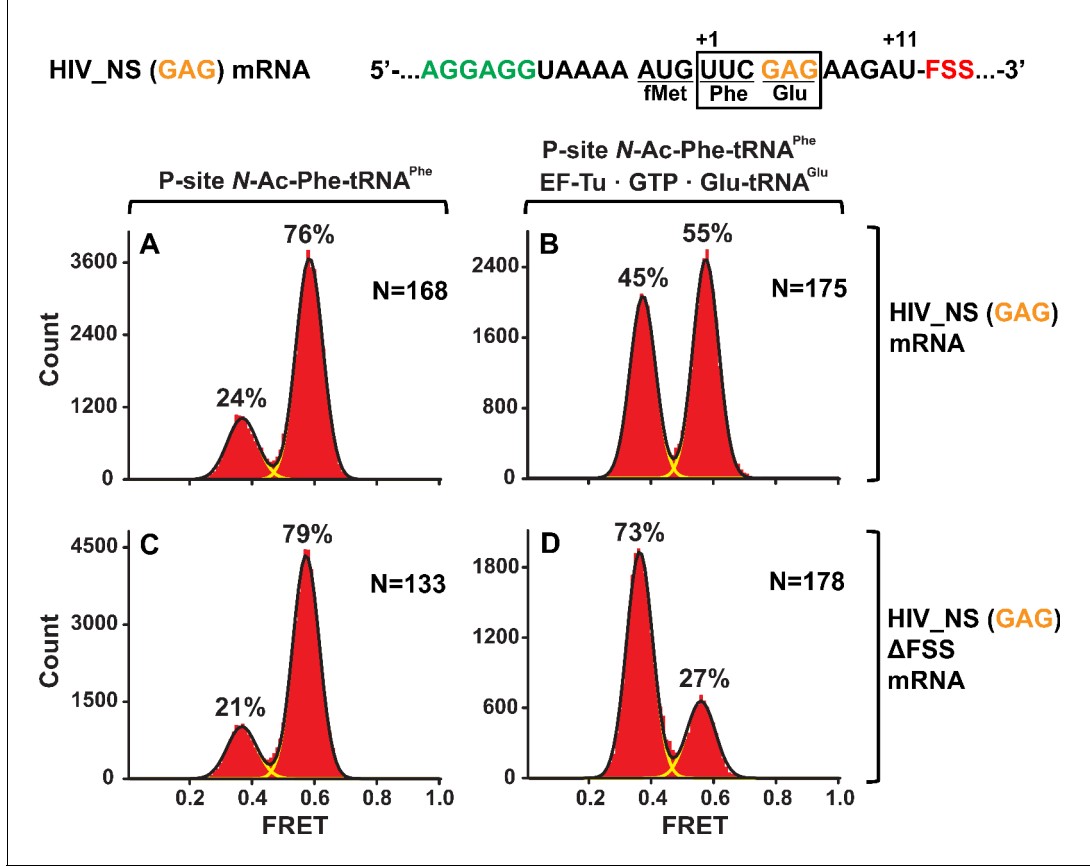

**Figure 5.** The FSS-induced ribosome stalling in NR conformation is independent of A-site codon identity. Histograms show FRET distributions in S6-cy5/L9-cy3 ribosomes programmed with HIV_NS (GAG) (**A–B**) or HIV_NS (GAG) ΔFSS (**C–D**) mRNA, respectively. Ribosomes containing P-site *N*-Ac-Phe-tRNA$^{Phe}$ (**A, C**) were incubated with EF-Tu•GTP•Tyr-tRNA$^{Tyr}$ (**B, D**) for 5 min and imaged after removal of unbound aminoacyl-tRNAs. Yellow lines show individual Gaussian fits of FRET distributions. Black lines indicate the sum of Gaussian fits. N indicates the number of FRET traces compiled into each histogram. The fractions of the ribosome in R and NR conformations are shown above the corresponding 0.4 and 0.6 Gaussian peaks, respectively. The online version of this article includes the following figure supplement(s) for figure 5:

**Figure supplement 1.** In the absence of slippery sequence, levels of the FSS-induced frameshifting are negligible.

**Figure supplement 2.** The FSSs do not stabilize the pretranslocation ribosome in the NR conformation.

(0.6 FRET to 0.4 FRET), respectively, and spent 80% of time in the R conformation (*Figure 5—figure supplement 2A–D*). Hence, neither dnaX FSS nor HIV FSS placed near the entry of the mRNA channel directly affect intersubunit dynamics. dnaX and HIV FSSs also did not change the sensitivity of P-site *N*-Ac-Phe-tRNA$^{Phe}$ toward the A-site aminoacyl-tRNA mimic, antibiotic puromycin (*Figure 6—figure supplement 1A*), indicating that the frameshifting-inducing stem-loops placed at the entry of the mRNA channel do not block the transpeptidase activity of the ribosome. Therefore, in our smFRET experiments, FSSs from dnaX and HIV likely stall the ribosome in the NR conformation by inhibiting A-site tRNA binding.

## dnaX and HIV FSS inhibit tRNA binding to the A site of the ribosome

To further test whether FSSs positioned near the entry of the mRNA channel inhibit tRNA binding, we used a filter-binding assay to measure binding of radio-labeled aa-tRNA during translation through four (Met, Val, Phe and Tyr) consecutive codons along the dnaX_NS mRNA. The ribosomes containing P-site *N*-Ac-Met-tRNA$^{Met}$ were then incubated with EF-G•GTP, EF-Tu•GTP, Val-tRNA$^{Val}$, Phe-tRNA$^{Phe}$ and Tyr-tRNA$^{Tyr}$ before loading ribosomes onto a nitrocellulose filter and washing away unbound aa-tRNA. The experiment was repeated three times with one of the three aminoacyl-tRNAs radio labeled, that is using [$^{14}$C]Val-tRNA$^{Val}$, [$^{3}$H]Phe-tRNA$^{Phe}$ or [$^{3}$H]Tyr-tRNA$^{Tyr}$. Similar

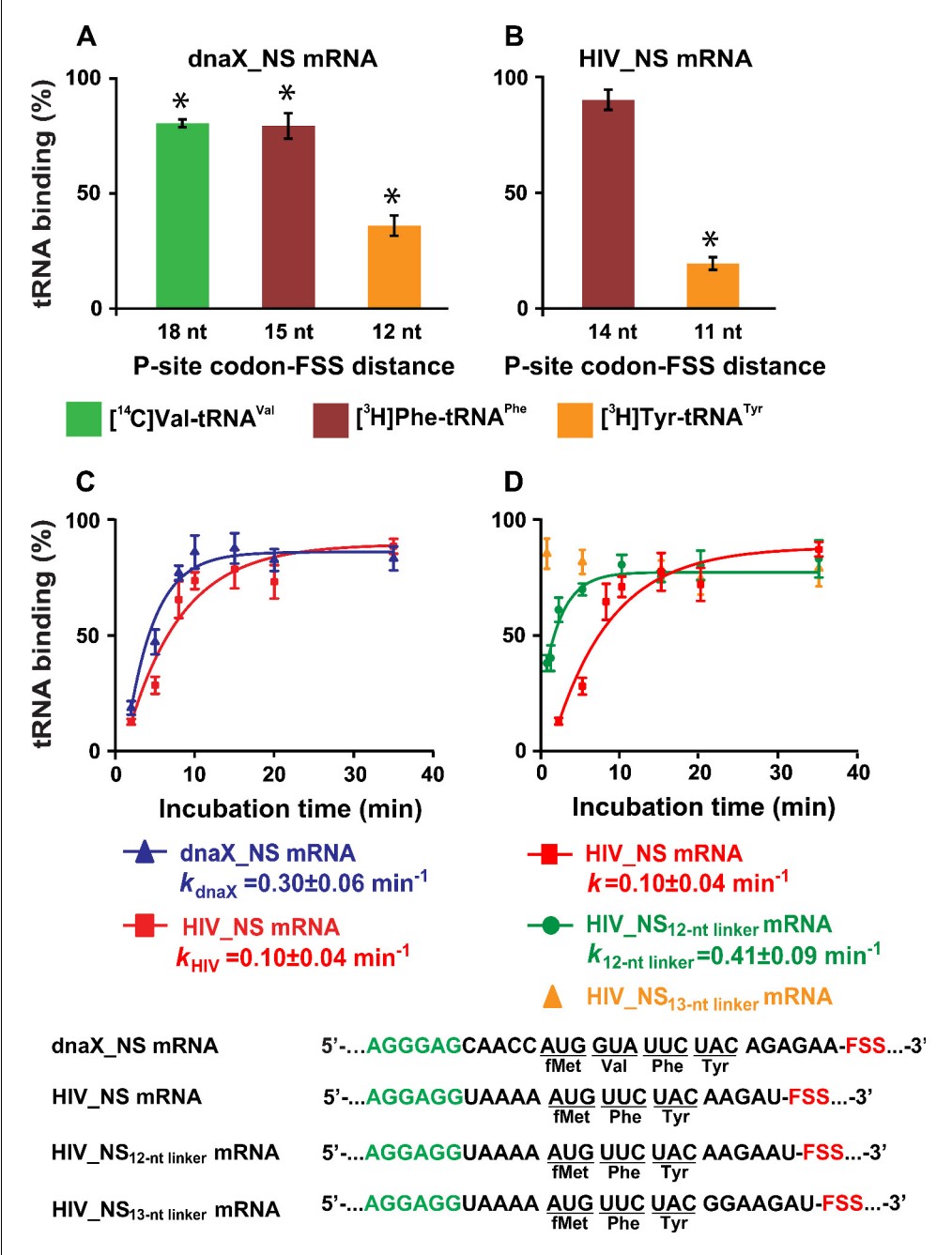

**Figure 6.** The dnaX and HIV FSSs inhibit A-site tRNA binding. (**A–B**) Incorporation of radio-labeled amino acids during translation through first four codons of dnaX_NS mRNA (**A**) or first three codons of HIV_NS mRNA (**B**) were measured by filter-binding assays. (**C**) Kinetics of EF-Tu-catalyzed [³H] Tyr-tRNA$^{Tyr}$ binding to the A site of ribosomes containing *N*-Ac-Phe-tRNA$^{Phe}$ in the P site. Ribosomes were programmed with dnaX_NS mRNA (blue) or HIV_NS mRNA (red). Single exponential fits are shown as line graphs. (**D**) Kinetics of EF-Tu-catalyzed [³H]Tyr-tRNA$^{Tyr}$ binding to the A site of ribosomes containing *N*-Ac-Phe-tRNA$^{Phe}$ in the P site. Ribosomes were programmed with HIV_NS mRNA (red), HIV_NS$_{12\text{-nt linker}}$ mRNA (green), and HIV_NS$_{13\text{-nt linker}}$ mRNA (yellow), respectively. The binding of radio-labeled amino acids to ribosomes programmed with FSS-containing mRNA is shown relative to that observed in ribosomes programmed with corresponding ΔFSS mRNA (**A–D**). Asterisks indicate that amino acid incorporation into ribosomes programmed with FSS-containing mRNA was significantly different from that in ribosomes programmed with ΔFSS mRNA, as *p*-values determined by the Student t-test were below 0.05. Error bars in each panel show standard deviations of triplicated measurements.

The online version of this article includes the following figure supplement(s) for figure 6:

**Figure supplement 1.** The FSS inhibits A-site tRNA binding but not ribosome-catalyzed transpeptidation reaction.

experiments were performed with ribosomes programmed with HIV_NS mRNA to measure binding of radio-labeled aa-tRNA during translation through three (Met, Phe and Tyr) consecutive codons.

In ribosomes programmed with dnaX_NS mRNA, binding of [$^3$H]Tyr-tRNA$^{Tyr}$ was considerably diminished while the incorporation of Val and Phe into the polypeptide chain were only mildly inhibited compared to levels measured in ribosomes programmed with dnaX_NS ΔFSS mRNA lacking the FSS (*Figure 6A*). Likewise, in ribosomes programmed with HIV_NS mRNA, binding of [$^3$H]Tyr-tRNA-$^{Tyr}$ was strongly inhibited while binding [$^3$H]Phe-tRNA$^{Phe}$ was unaffected (*Figure 6B*).

Next, we examined the kinetics of [$^3$H]Tyr-tRNA$^{Tyr}$ binding to the A site of ribosomes, which contained P-site *N*-Ac-Phe-tRNA$^{Phe}$ and were programmed with either dnaX_NS or HIV_NS mRNA. Both dnaX and HIV FSSs dramatically slowed the rate of [$^3$H]Tyr-tRNA$^{Tyr}$ binding as the apparent pseudo first order rate of tRNA binding was reduced to 0.3 and 0.1 min$^{-1}$, respectively (*Figure 6C*). When ribosomes were programmed with either dnaX_NS ΔFSS or HIV_NS ΔFSS mRNAs, the rate of [$^3$H]Tyr-tRNA$^{Tyr}$ binding was too fast to be measured by the filter-binding assay, which involves manual mixing of ribosomes and tRNA. These kinetic experiments also show that while dnaX and HIV FSSs strongly inhibit A-site tRNA binding, they do not completely block it.

To further investigate how kinetics of tRNA binding depends on the spacing between the P-site codon and the FSS, we made two HIV_NS mRNA variants where the spacer between the P-site (UUC) codon and the FSS was extended from 11 (HIV_NS mRNA) to 12 (HIV_NS$_{12-nt\ linker}$ mRNA) or 13 nucleotides (HIV_NS$_{13-nt\ linker}$ mRNA), respectively. Extending the P-site-FSS spacer from 11 to 12 nucleotides increased the apparent pseudo first order rate of [$^3$H]Tyr-tRNA$^{Tyr}$ binding to the A site of the ribosome from 0.1 to 0.4 min$^{-1}$ (*Figure 6D*), however A-site binding remained severely inhibited relative to the ΔFSS control. By contrast, lengthening the P-site-FSS spacer to 13 nucleotide relieved the FFS-induced inhibition of A-site binding as the rate of [$^3$H]Tyr-tRNA$^{Tyr}$ binding became too fast to be measured by manual mixing kinetic measurements (*Figure 6D*). Hence, consistent with smFRET experiments, the filter-binding assay demonstrates that both dnaX and HIV FSSs inhibit A-site tRNA binding only when positioned 11 or 12 nucleotides downstream of the P-site codon.

Additionally, we found that in the absence of EF-Tu, FSSs from dnaX and HIV also inhibit non-enzymatic binding of *N*-Ac-[$^3$H]Tyr-tRNA$^{Tyr}$ to the A site of ribosomes programmed with dnaX_NS mRNA or HIV_NS mRNA (*Figure 6—figure supplement 1B*). Therefore, the FSS-induced inhibition of A-site tRNA binding is independent of the EF-Tu function. In addition, we observed that FSS-induced inhibition of cognate aminoacyl-tRNA binding to the A site is largely independent of identities of either A- or P-site codons (*Figure 7*). FSS-induced binding inhibition of cognate aminoacyl-tRNA to the A site was observed when filter-binding experiments were performed with ribosomes programmed with mRNAs in which the original UAC (Tyr) or UUC (Phe) codons of HIV_NS mRNA were replaced with the GAG (Glu) or AUG (Met) codon, respectively (*Supplementary file 1*).

Taken together, our smFRET and filter-binding experiments indicate that when positioned 11–12 nucleotides downstream of the first nucleotide of the P-site codon, the FSSs from HIV and *dnaX* mRNAs can substantially inhibit binding of aminoacyl-tRNA to the A site of the ribosome. Assuming that *dnaX* and HIV mRNA are threaded through the 30S mRNA channel, an 11–12 nucleotide distance from the P-site codon corresponds to positioning of the FSSs at the entry of the mRNA channel. Consistent with this hypothesis, a recent cryo-EM reconstruction revealed that the *dnaX* FSS placed 12 nucleotides downstream from the P-site codon interacts with ribosomal proteins uS3, uS4, and uS5 located at the 30S mRNA entry channel (*Zhang et al., 2018*). However, the mRNA entry channel is ~20 Å away from the 30S decoding center. How the FSS positioned at the mRNA entry channel inhibits tRNA binding to the A site remains unclear.

## Cryo-EM analysis reveals HIV FSS hairpin binding to the A site

To investigate the structural basis for the tRNA binding inhibition by the FSSs, we performed single-particle cryo-EM of the HIV FSS mRNA-ribosome complex. We prepared a 70S *E. coli* ribosome complex programmed with HIV_NS (GAG) mRNA (*Figure 1*) and bound with a peptidyl-tRNA analog, *N*-Ac-Phe-tRNA$^{Phe}$, in the P site. Our smFRET and filter-binding experiments showed that in this complex, the HIV FSS inhibits binding of Glu-tRNA$^{Glu}$ to the Glu (GAG) codon in the A site (*Figures 5* and *7*).

Maximum-likelihood classification of a 640,261-particle data set revealed predominant ribosome states that contained strong density in both the P and A sites, which we interpreted as P-site tRNA and the FSS hairpin, respectively (64% particles total) (*Figure 8—figure supplement 1*). Two classes

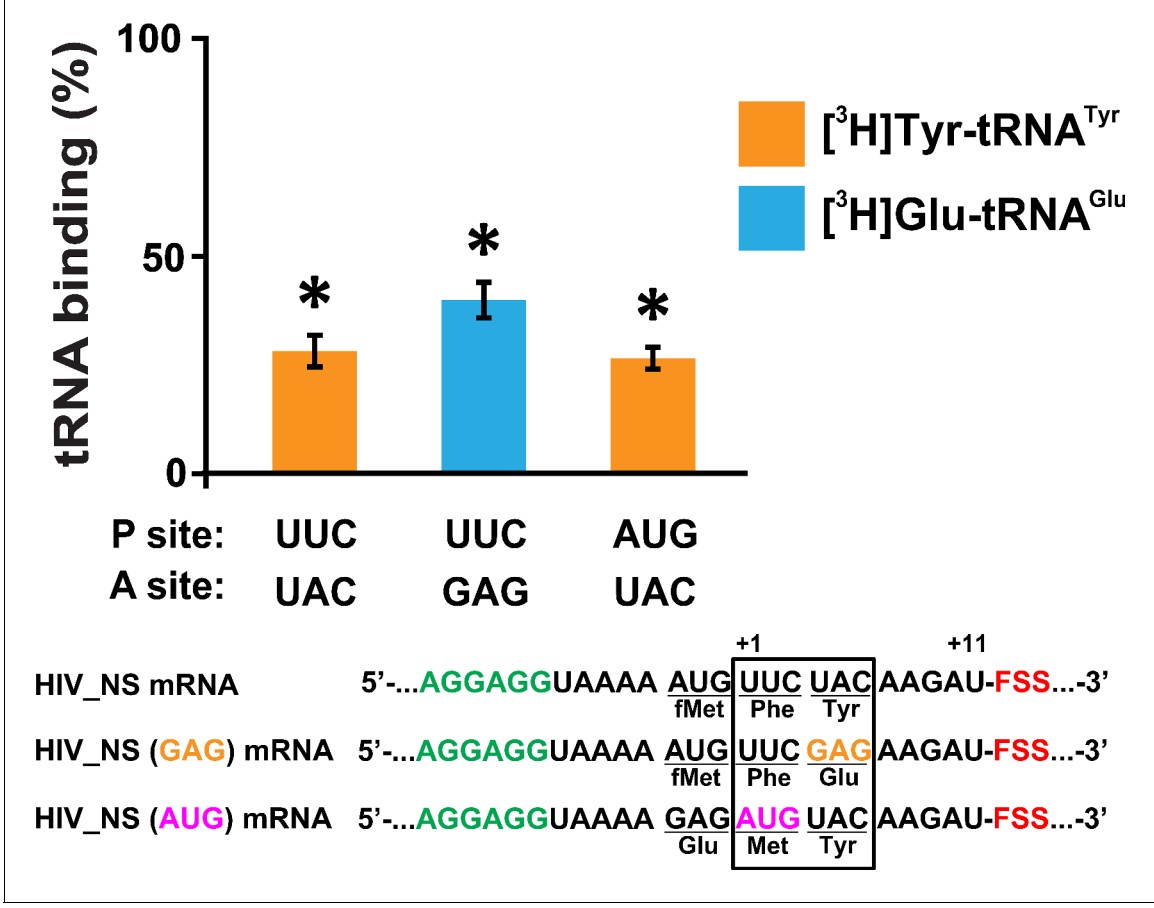

**Figure 7.** The FSS-mediated inhibition of A-site tRNA binding is independent of P-site and A-site codon identities. The extent of EF-Tu-catalyzed cognate aminoacyl-tRNA binding after a 5 min incubation with ribosomes programmed with HIV_NS, HIV_NS (GAG) or HIV_NS (AUG) mRNAs. The P site of the ribosome was bound with N-Ac-Phe-tRNA$^{Phe}$ (in the presence of HIV_NS and HIV_NS (GAG) mRNAs) or N-Ac-Met-tRNA$^{Met}$ (in the presence of HIV_NS (AUG) mRNA). The binding of radio-labeled amino acids to ribosomes programmed with FSS-containing mRNA is shown relative to that observed in ribosomes programmed with corresponding ΔFSS mRNA. Asterisks indicate that amino acid incorporation into ribosomes programmed with FSS-containing mRNA was significantly different from that in ribosomes programmed with ΔFSS mRNA, as p-values determined by the Student t-test were below 0.05. Error bars in each panel show standard deviations of triplicated measurements.

comprise the ribosome in classical NR states (NR-I and NR-II,~1˚ 30S rotation) with P/P tRNA (*Figure 8A*, *Figure 8—figure supplement 2A*), while one class represents a R ribosome state with P/E tRNA (R-I,~7˚ 30S rotation) (*Figure 8B*, *Figure 8—figure supplement 2B*), at overall resolutions between 3.1 Å and 3.4 Å. Additional classes contained weaker density in the A site, likely reflecting compositional and/or conformational heterogeneity (see Materials and methods). By contrast, there is no density at the entry site of the mRNA channel, indicating that the HIV FSS does not bind to the mRNA entry channel.

P-site tRNA is base paired with mRNA, indicating the absence of frameshifting. In all three classes NR-I, NR-II and R-I, density allows for the distinction of purines from pyrimidines (*Figure 8C,D*), revealing Watson-Crick pairing between N-Ac-Phe-tRNA$^{Phe}$ and an in-frame UUC codon. This is consistent with smFRET data showing that in the absence of the slippery sequence, *dnaX* and HIV FSSs do not promote frameshifting (*Figure 5—figure supplement 1*).

In both NR-I and NR-II structures, the P-site tRNA elbows are shifted by 11.8 Å and 13.0 Å towards the E site compared to P-site tRNA in other classical states (corresponding to tRNA rotation by 18.5˚ and 21.0˚, respectively compared to PDBID: 4V5D, *Figure 8E*). Similar tRNA states were observed in several termination complexes where they are thought to represent an intermediate to the P/E hybrid-state after deacylation by a release factor (*Graf et al., 2018*; *Svidritskiy et al., 2019*; *Figure 8—figure supplement 2C*). The peptidyl moiety is unresolved in both NR-I and NR-II

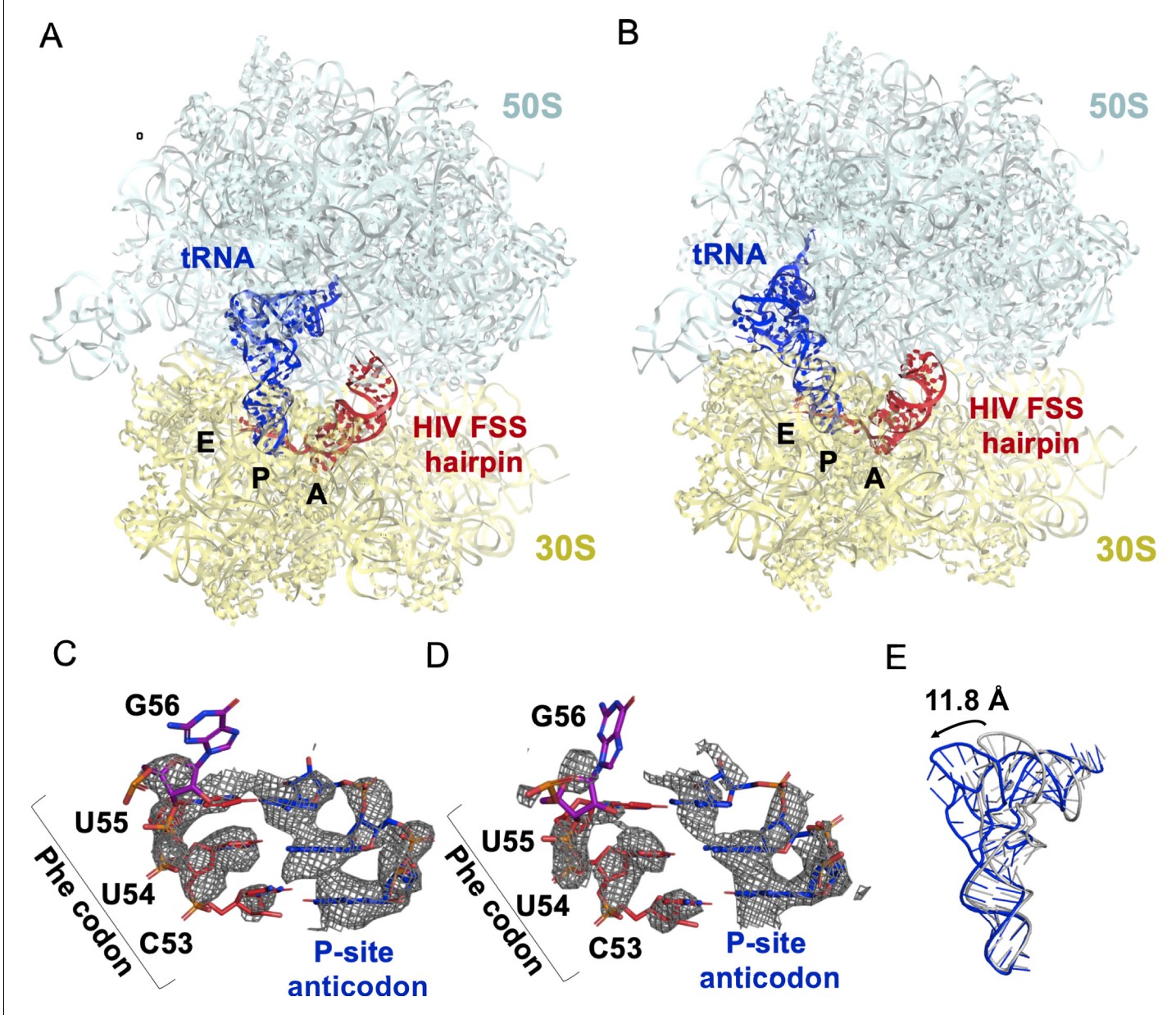

**Figure 8.** The HIV FSS hairpin occupies the ribosomal A site. (**A**) Cryo-EM structures of the 70S ribosome in non-rotated (NR-I) and (**B**) rotated (R–I) conformations. The large subunits (50S) are shown in aqua, the small subunits (30S) in yellow, P-site tRNA in blue, and HIV FSS hairpin in red. (**C** and **D**) Close-up views of the codon and anti-codon basepairs of the NR-I (**C**) and R-I (**D**) states illustrating in-frame basepairing of the HIV_NS(GAG) mRNA (red) with the P-site tRNA (blue). The first position of the GAG A-site codon is shown in purple. The cryo-EM map (gray mesh) was sharpened by applying a B-factor of $-50$ Å$^2$. (**E**) Overlay of NR-I P-site tRNA with P-site tRNA bound in the P/P classical site (PDB ID: 4V5D) shows a 11.8 Å rotation of the tRNA elbow towards the E site.

The online version of this article includes the following figure supplement(s) for figure 8:

**Figure supplement 1.** Data and schematic of cryo-EM refinement and classification.
**Figure supplement 2.** Density maps of (**A**) NR-I and (**B**) R-I filtered to 5 Å resolution.

structures, so it is unclear whether the peptidyl moiety is disordered or hydrolyzed leading to deacylation of the P-site tRNA. Nevertheless, the position of the tRNA CCA tail in the 50S peptidyl-transferase center is similar to that in the peptidyl-tRNA complexes (*Polikanov et al., 2014*), suggesting that NR-I and NR-II can be sampled with peptidyl-tRNAs. Furthermore, superposition with the P/P-

tRNA bound structures demonstrates the absence of steric clash between the hairpin and tRNA, indicating that the hairpin in the A site is also compatible with the classical NR ribosome.

Our cryo-EM reconstructions reveal the structural basis for FSS binding to the ribosome. Instead of binding next to the mRNA channel, the FSS hairpin stacks on the purine-rich GAG codon and nucleotides AAGAU (nucleotides 59–63 of HIV (GAG) NS mRNA between GAG and the FSS), which we further refer to as the linker sequence, to occupy the ribosomal A site (*Figure 9A–C*). To exclude the possibility that the A-site density is an unusually accommodated A-site tRNA, we docked a tRNA into this density. In both the R and NR states, no additional density is observed that could correspond to the acceptor stem of A-site tRNA. Furthermore, extended portions of the tRNA elbow and acceptor stem would clash with the A-site finger of the 50S subunit (*Figure 9A*, *Figure 8—figure supplement 2D,E*) because the density is rotated by ~15° towards the A-site finger (NR-I) compared to A-site tRNA in the classical state (*Figure 9A*). In the R-I conformation, the hairpin density merges

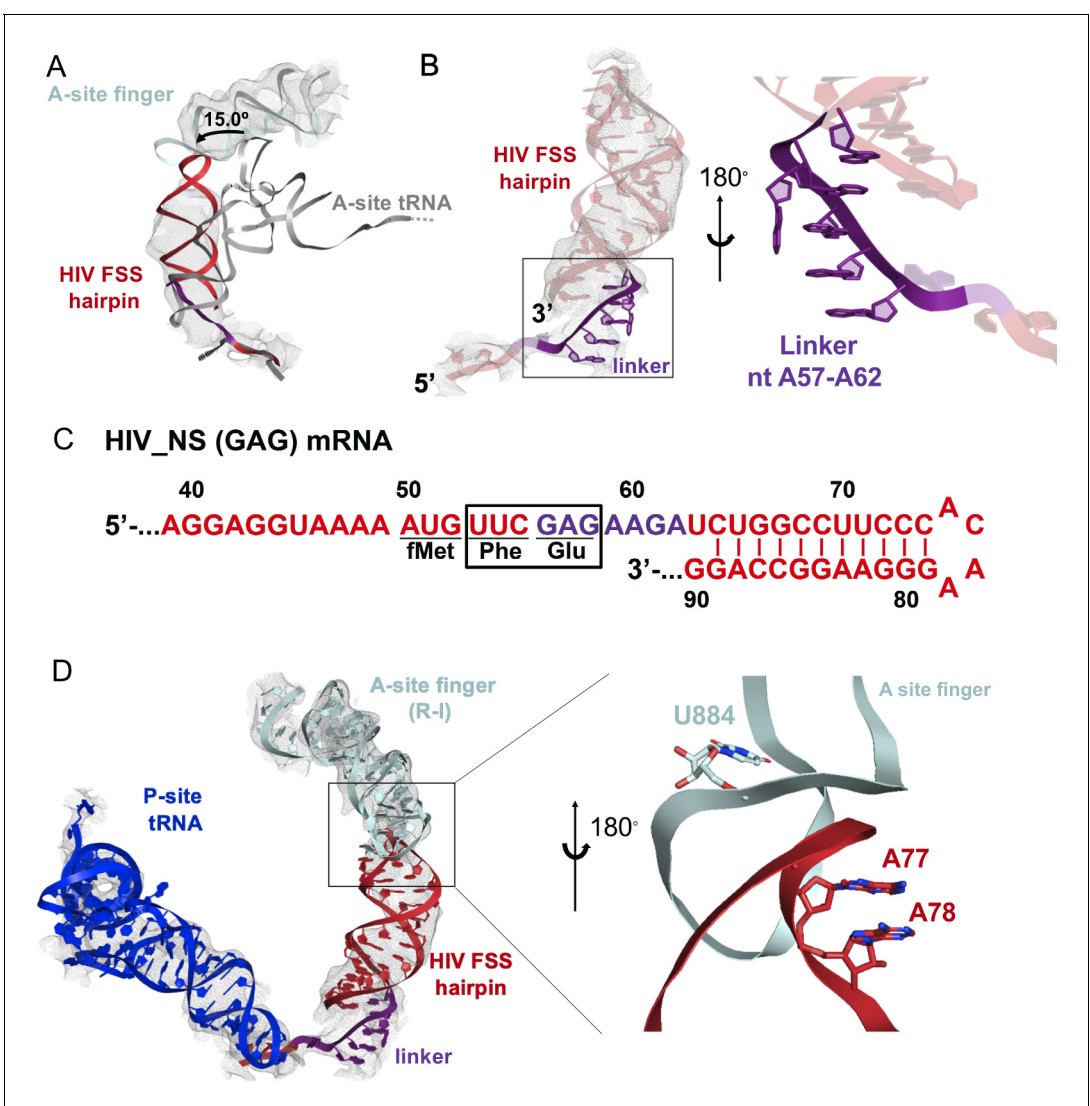

**Figure 9.** Stacking interactions of linker nucleotides stabilize the HIV FSS in the A site. (**A**) Overlay of the NR-I HIV FSS hairpin from this work with A-site tRNA (gray) accommodated in a ribosome in the classical state (PDB ID: 4V5D). The hairpin density is shown after filtering to 8 Å. (**B**) View of the R-I HIV FSS hairpin model (red, linker in purple) in cryo-EM density filtered to 5 Å (gray mesh) and close-up of the purine stack (shown in purple) after 180° rotation. (**C**) Primary sequence and secondary structure of the HIV_NS(GAG) mRNA. The linker sequence is highlighted in purple. (**D**) In the rotated state, the HIV FSS hairpin (red) contacts the A-site finger (aqua) of the large ribosomal subunit. The hairpin density allows to clearly identify helical pitch, major and minor grooves of the A-form RNA. The close-up after 180° rotation shows that the only complementary bases within the two loop regions point away from each other and likely do not contribute to A-site finger/hairpin binding. P-site tRNA is blue.

with the A-site finger (helix38) density suggesting that the hairpin loop (A75-A78) comes into a closer contact with the A-site finger (helix 38) (*Figure 9D*) and does not correspond to a hybrid-state A/P tRNA. Thus, our interpretation of the density rules out tRNA in A site and explains inhibition of A-site tRNA binding by the presence of the FSS hairpin.

The local resolution of the A site allows to distinguish the major and minor grooves of the A-form RNA, but individual basepair locations cannot be determined, suggesting that the FSS hairpin samples an ensemble of conformations. We were able to build a plausible pseudoatomic model of the linker and hairpin based on resolved density features and previously solved structures. We used a published NMR model of the HIV FSS hairpin (*Staple and Butcher, 2003*), which includes the hairpin loop and adjacent nine basepairs (nucleotides 66–87) of the predicted 11 canonical and additional G-U pair (predicted using mfold [*Zuker, 2003*]). The closing hairpin basepairs, A-site codon (GAG) and the linker sequence were modeled manually to form stacking interactions. The density suggests a dynamic conformation of the first A-site codon position (G56). The following six nucleotides (A57-A62, sequence: AGAAGA) contain an adenine-rich homopurine sequence, which typically adopts stacked A-form-like conformations (*Isaksson et al., 2004*). Accordingly, we modeled the second and third positions of the A-site codon and the linker as an A form-like purine stack (*Figure 9B*). A kink in the density suggests that one nucleotide in the purine stack, possibly A60, is flipped out of the helix. Nucleotides G61 and A62 might form heteropurine basepairs with A92 and G91, respectively, though specific interactions are not visible at this resolution.

Conformational heterogeneity of the hairpin observed in our cryo-EM structures is likely important for the hairpin's mechanism of action. To allow FSS hairpin entry into the A site, the mRNA linker has to be released from the mRNA channel in the 30S subunit. Dynamic occupancy of the A site by the FSS hairpin may allow an incoming tRNA to eventually overcome the translational block that the hairpin imposes on the ribosome. Binding of tRNA likely stabilizes the A-site codon, allowing the linker to reestablish its position in the mRNA channel and the hairpin to bind next to the channel entry, restoring an elongation ribosome state.

## Discussion

In this study, we investigated molecular mechanisms by which FSSs from *E. coli dnaX* and HIV mRNAs induce ribosome pausing. Although the sequences of these mRNA elements are different, they appear to act via the same mechanism due to similar hairpin structures. We found that when positioned 11–12 nucleotides downstream of the P-site codon, the FSSs perturb translation elongation through two parallel pathways: (i) inhibiting tRNA binding to the A site of the ribosome and (ii) inhibiting ribosome translocation (*Figure 10*). These observations support the idea that FSSs stimulate frameshifting by pausing the ribosome. Our finding that the dnaX FSS slows the rate of ribosome translocation by ~10 fold is also in agreement with a number of previous reports (*Caliskan et al., 2017*; *Chen et al., 2014*; *Choi et al., 2020*; *Kim et al., 2014*; *Kim and Tinoco, 2017*).

While dnaX and HIV FSSs dramatically perturb the kinetics of the elongation cycle, our data provide no evidence that these stem-loops induce a unique conformation of the ribosome with a 'super-rotated' orientation of ribosomal subunits reported previously (*Qin et al., 2014*). The super-rotated conformation, in which the ribosomal 30S subunit rotates by ~20 degrees against the 50S subunit, was inferred from smFRET data showing a 0.2 FRET value for the S6/L9 FRET pair when the ribosome encountered a dnaX FSS or mRNA/DNA duplex. In our smFRET study using ribosomes programmed with dnaX_Slip mRNA, we only detected fluctuations between 0.4 and 0.6 FRET states corresponding to R and NR conformations while no FRET states below 0.4 were observed. Previously observed 0.2 FRET of the S6/L9 FRET pair might correspond to nuclease- or protease-damaged ribosomes, or they are induced by the interaction of ribosomes with the microscope slide surface.

In our work, we observed a 5-fold decrease of the rate of Lys-tRNA[Lys] binding to the second Lys codon of the dnaX slippery sequence induced by the FSS (*Figure 2*). Furthermore, the rate of tRNA binding decreased by another order of magnitude when the slippery sequence of *dnaX* mRNA was replaced with non-slippery codons (*Figures 3* and *6*). A-site tRNA inhibition was also observed with the HIV FSS in the context non-slippery codons (*Figures 4*, *5* and *7*), suggesting a common underlying mechanism employed by a variety of FSSs. Such inhibition is only observed when the FSS is positioned 11–12 nucleotides downstream of the P-site codon.

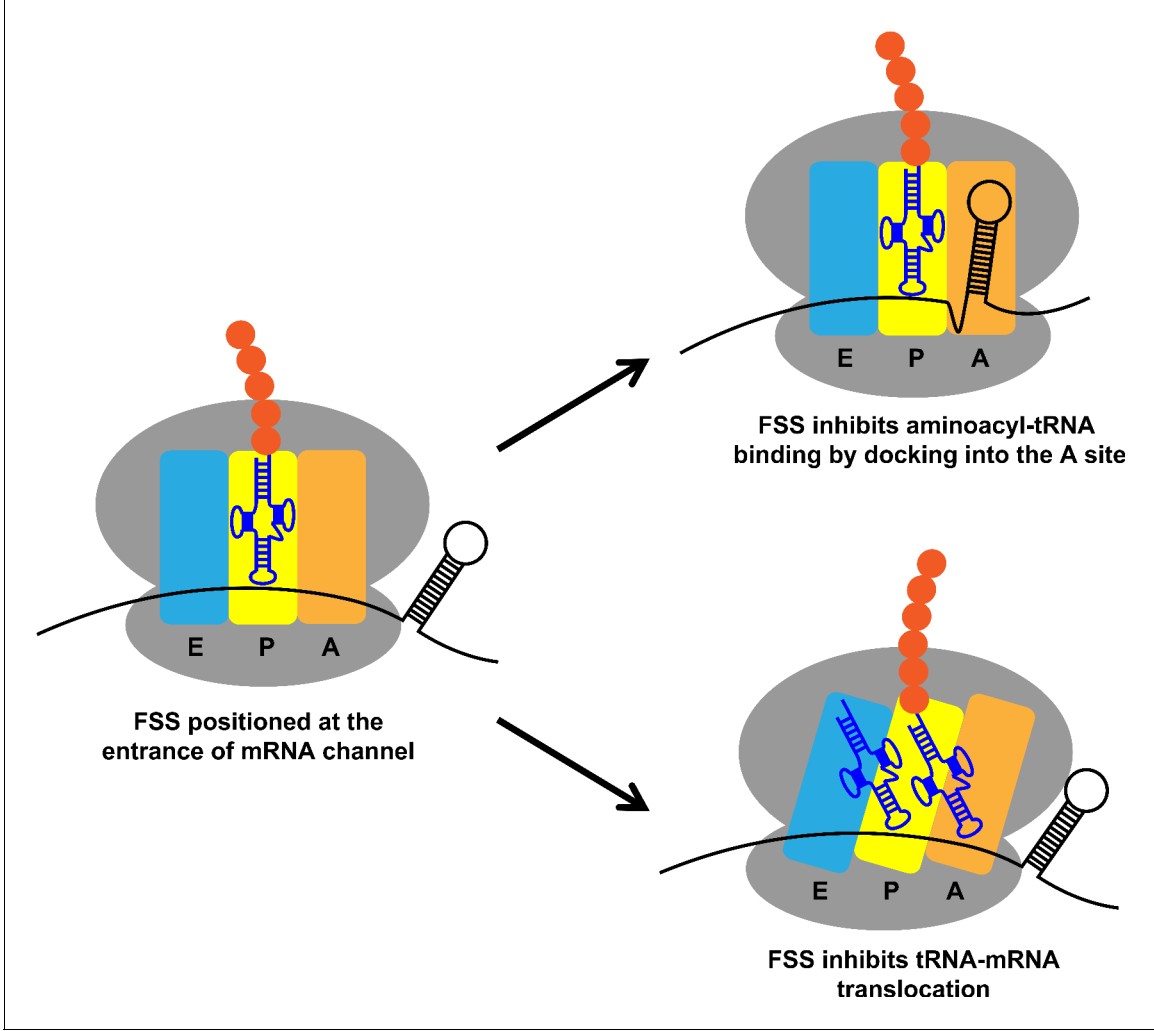

**Figure 10.** Two parallel mechanisms by which *dnaX* and HIV FSSs perturb translation elongation. Upon encountering the ribosome, the FSS can hinder tRNA binding by docking to the A site of the ribosome or inhibit translocation by interacting with the mRNA entry channel.

Our observation of FSS-dependent inhibition of A-site tRNA binding helps to resolve some inconsistencies in previous studies. One previous smFRET study suggests that the rate of A-site tRNA delivery during decoding of the slippery sequence of *dnaX* mRNA is unaffected by the presence of downstream FSS (*Kim et al., 2014*). Other single-molecule experiments suggest that Lys-tRNA$^{Lys}$ accommodation during decoding of the second Lys codon of dnaX slippery sequence is delayed (*Chen et al., 2014*). Two-fold inhibition of A-site codon decoding induced by the FSS was observed in another study when the dnaX slippery sequence was replaced with non-slippery Lys codons AAGAAG (*Caliskan et al., 2017*). Differences in experimental conditions (EF-Tu and tRNA concentrations), as well as sequence variations of model dnaX mRNAs may underlie inconsistencies between earlier studies of the FSS effect on tRNA binding. In particular, in the aforementioned studies, the wild-type purine-rich dnaX linker sequence AGUGA was replaced with UUUGA (*Kim et al., 2014*), UUCUA (*Caliskan et al., 2017*) or AGUUC (*Chen et al., 2014*).

When positioned 11–12 nucleotides downstream of the P-site codon, FSSs likely inhibit A-site tRNA binding and ribosome translocation by sampling two alternative conformations on the ribosome (*Figure 10*), consistent with our observation of conformational heterogeneity of the hairpin in the A site. In one FSS conformation, which was previously seen in a cryo-EM reconstruction of dnaX-ribosome complex (*Zhang et al., 2018*), the FSS interacts with the mRNA entry channel. It has been recently demonstrated that upon encountering mRNA secondary structure the ribosome translocates through two alternative (fast and slow) pathways (*Desai et al., 2019*). The interactions of FSSs with

the mRNA entry channel may increase the flux through the slow pathway and thus decrease the average rate of ribosome translocation (*Desai et al., 2019*). In another FSS conformation, which was visualized by our single-particle cryo-EM analysis, nucleotides between the P site and the FSS disassociate from the mRNA channel, and the HIV FSS docks into the A site thus sterically hindering tRNA binding.

The linker sequence between the A-site codon and the FSS may facilitate binding of the FSS into the A site. Our structure suggests that the homo-purine sequence encompassing the second and third positions of the A-site codon and the linker form a purine stack. The formation of an A-form like single-stranded helix by the linker nucleotides may make the release of the mRNA nucleotides between P-site codon and FSS from the mRNA channel and docking of the FSS into the A site more thermodynamically favorable.

The HIV FSS is the largest but not the first hairpin observed in the ribosomal A site. A short four-basepair-long hairpin was observed in the A site of the ribosome bound to a short model mRNA (MF36) derived from phage T4 gene 32 mRNA (*Yusupova et al., 2001*). Crystallographic analyses suggested that the hairpin following the start codon may facilitate initiation of translation of gene 32 (*Yusupova et al., 2001*). Another short five-basepair-long hairpin of the bacteriophage T4 gene 60 was visualized by cryo-EM in the A-site of the ribosome, which was stalled at the 'take-off' mRNA site containing a stop codon (*Agirrezabala et al., 2017*). The hairpin prevents binding of the release factor 1 (RF1), thus inhibiting translation termination and inducing translational bypassing (*Agirrezabala et al., 2017*). Another short two-basepair-long hairpin was observed in the A site of the ribosome in which the P and A sites were occupied by an inhibitory codon pair CGA-CCG that is known to cause ribosome stalling (*Gamble et al., 2016*; *Tesina et al., 2020*). These observations suggest that occlusion of the A site by RNA stem-loops may be a common strategy shared by many regulatory stem-loops.

Our findings provide new insights into the mechanisms of −1 PRF stimulated by stem-loop FSSs. Several lines of evidence suggest that −1 PRF can occur through three different pathways: (i) slippage of the single P-site tRNA when the A site remains vacant, (ii) frameshifting of both A-site and P-site tRNAs during aa-tRNA accommodation to the A site, and (iii) slippage of A- and P-site tRNAs during translocation (*Dinman, 2012*). A number of studies indicated that −1 PRF on both *dnaX* and HIV mRNAs involves P-site tRNA slippage and frameshifting during translocation of two tRNAs, that is pathways (i) and (iii) (*Brunelle et al., 1999*; *Caliskan et al., 2017*; *Jacks et al., 1988*; *Korniy et al., 2019a*; *Léger et al., 2007*; *Yelverton et al., 1994*). Partitioning between these pathways is modulated by tRNA abundance (*Caliskan et al., 2017*; *Korniy et al., 2019a*; *Korniy et al., 2019b*). Our observation that frameshift-inducing hairpins inhibit both A-site tRNA binding and mRNA translocation are consistent with single tRNA slippage (i) and translocation (iii) frameshifting pathways, respectively. It is not clear whether pseudoknot FSSs also utilize both pathways or A-site inhibition/P-site tRNA slippage is unique to stem-loop FSSs.

Our findings of hairpin competition with tRNA expose a novel mechanism that stem-loop FSSs in retroviruses, including HIV, likely employ to regulate viral gene expression and expand the viral proteome via mRNA frameshifting. Likewise, transient binding of FSS stem-loops to the A site of the ribosome may mediate frameshifting in insertion sequences of the IS1-IS3 family of bacterial transposable elements (*Chandler and Fayet, 1993*) and orf 1a/1b of astroviruses (*Marczinke et al., 1994*). Transient occlusion of the A site by an mRNA hairpin may also underlie programmed ribosome pausing/stalling events that trigger targeting a nascent polypeptide chain to a membrane (*Young and Andrews, 1996*) and No-Go mRNA decay (*Doma and Parker, 2006*).

## Materials and methods

**Key resources table**

| Reagent type (species) or resource | Designation | Source or reference | Identifiers | Additional information |
|---|---|---|---|---|
| Gene (*Escherichia coli*) | *dnaX* | doi: 10.1093/nar/14.20.8091 | Uniprot ID: P06710 | |

*Continued on next page*

*Continued*

| Reagent type (species) or resource | Designation | Source or reference | Identifiers | Additional information |
|---|---|---|---|---|
| Gene (Human Immuno deficiency Virus Type 1) | *gag-pol* | doi: 10.1089/aid.1987.3.57 | Uniprot ID: P04585 | |
| Strain, strain background (*Escherichia coli*) | MRE600 | ATCC | ATCC #29417, (NCTC #8164, NCIB #10115) | *E. coli* strain K-12 that lacks the RNase I activity |
| Strain, strain background (*Escherichia coli*) | DH5α competent cell | Thermo Fisher Scientific | Catalog #: 18265017 | |
| Genetic reagent (*Escherichia coli*) | tRNA | Chemical Block | tRNA$^{Phe}$ tRNA$^{Tyr}$ tRNA$^{fMet}$ tRNA$^{Met}$ tRNA$^{Glu}$ tRNA$^{Val}$ tRNA$^{Lys}$ tRNA$^{Arg}$ | |
| Genetic reagent (*Escherichia coli*) | Total tRNA from *E. coli* MRE600 | Sigma-Aldrich | Catalog #: 10109541001 | |
| Transfected construct (*Escherichia coli*) | pSP64 poly (A) | Promega | Catalog #: P1241 | |
| Biological sample (*Escherichia coli*) | ribosome (30S, 50S and 70S) | doi: 10.1016/j.jmb.2007.04.042 doi: 10.1073/pnas.1520337112 | | |
| Recombinant DNA reagent | SacI-HF | New England Biolabs | Catalog #: R3156 | |
| Recombinant DNA reagent | BglII | New England Biolabs | Catalog #: R0144 | |
| Recombinant DNA reagent | HindIII-HF | New England Biolabs | Catalog #: R3104 | |
| Recombinant DNA reagent | T4 DNA ligase | New England Biolabs | Catalog #: M0202 | |
| Peptide, recombinant protein | elongation factor Tu (EF-Tu) | doi: 10.1016/j.jmb.2007.04.042 | | |
| Peptide, recombinant protein | elongation factor G (EF-G) | doi: 10.1016/j.jmb.2007.04.042 | | |
| Peptide, recombinant protein | T7 polymerase | doi: 10.1073/pnas.95.2.515 | | |
| Commercial assay or kit | Plasmid Mini prep System | Promega | Catalog #: A1223 | |
| Commercial assay or kit | Gel and PCR Clean-Up System | Promega | Catalog #: A9281 | |
| Commercial assay or kit | DNA oligo synthesis | INTEGRATED DNA TECHNOLOGIES (IDT) | | |
| Commercial assay or kit | DNA sequencing | ACGT, INC | | |
| Chemical compound, drug | puromycin | Sigma-Aldrich | Catalog #: P8833 | |
| Chemical compound, drug | cy3 maleimide | Click Chemistry Tools | Catalog #: 1009 | |
| Chemical compound, drug | cy5 maleimide | Click Chemistry Tools | Catalog #: 1004 | |
| Chemical compound, drug | Phenylalanine, L -[2,3,4,5,6-3H]- | PerkinElmer | Catalog #: NET112201MC | |
| Chemical compound, drug | Valine, L-[U-$^{14}$C]- | PerkinElmer | Catalog #: NEC291EU050UC | |

*Continued on next page*

*Continued*

| Reagent type (species) or resource | Designation | Source or reference | Identifiers | Additional information |
|---|---|---|---|---|
| Chemical compound, drug | Methionine, L-[$^{35}$S]- | PerkinElmer | Catalog #: NEG009T001MC | |
| Chemical compound, drug | Glutamic Acid, L-[3,4–$^{3}$H]- | PerkinElmer | Catalog #: NET490001MC | |
| Chemical compound, drug | Tyrosine, L -[ring-3,5 | PerkinElmer | Catalog #: NET127001MC | |
| Software, algorithm | smFRET data acquisition and analysis package | Taekjip Ha's laboratory website at Johns Hopkins University (http://ha.med.jhmi.edu/resources/) | | |
| Software, algorithm | IDL | ITT, INC. (https://www.harrisgeospatial.com/Software-Technology/IDL) | | |
| Software, algorithm | HaMMy | Taekjip Ha's laboratory website at Johns Hopkins University (http://ha.med.jhmi.edu/resources/) doi: 10.1529/biophysj.106.082487 | | |
| Software, algorithm | SerialEM | (https://bio3d.colorado.edu/SerialEM/) doi: 10.1016/j.jsb.2005.07.007 | | |
| Software, algorithm | cisTEM | (https://cistem.org/) DOI: 10.7554/eLife.35383 | | |
| Software, algorithm | Phenix-1.17.1–3660 | (https://www.phenix-online.org/) DOI: 10.1107/S2059798319011471 | | |
| Software, algorithm | Coot v0.9 pre-EL | Part of CCPEM 1.3.0 suite (https://www.ccpem.ac.uk/index.php) DOI: 10.1107/S2059798317007859 | | |
| Software, algorithm | PyMol 2.3.2 | Schrödinger, LLC (https://pymol.org) | | |

## Ribosome, EF-G, EF-Tu and tRNA preparation

tRNA$^{fMet}$, tRNA$^{Met}$, tRNA$^{Phe}$, tRNA$^{Val}$, tRNA$^{Tyr}$, tRNA$^{Lys}$, and tRNA$^{Glu}$ (purchased from Chemical Block) were aminoacylated as previously described (*Lancaster and Noller, 2005*; *Moazed and Noller, 1989*). Tight couple 70S ribosomes used for biochemical experiments and ribosomal subunit used for cryo-EM sample assembly were purified from *E. coli* MRE600 stain as previously described (*Ermolenko et al., 2007*). S6-Cy5/L9-Cy3 ribosomes were prepared by partial reconstitution of ΔS6-30S and ΔL9-50S subunits with S6-41C-Cy5 and L11-11C-Cy3 as previously described (*Ermolenko et al., 2007*; *Ling and Ermolenko, 2015*). Histidine-tagged EF-G and EF-Tu were expressed and purified using previously established procedures (*Ermolenko et al., 2007*).

## Preparation of model mRNAs

Sequences encoding dnaX and HIV mRNAs were cloned by directional cloning downstream of T7 promoter in pSP64 plasmid vector (Promega Co). Model mRNAs (*Supplementary file 1*) were generated by T7 polymerase-catalyzed run-off in vitro transcription and purified by denaturing PAGE. Prior to transcription, 3' ends of the model mRNAs were defined by linearizing the corresponding DNA templates at specific restriction sites (*Supplementary file 1*). smFRET measurements smFRET measurements were done as previously described (*Cornish et al., 2008*; *Ling and Ermolenko, 2015*) with modifications. The quartz slides used for total internal reflection fluorescence (TIRF) microscopy were treated with dichlorodimethylsilane (DDS) (*Hua et al., 2014*). The DDS surface was coated with biotinylated BSA (bio-BSA). Uncoated areas were then passivated by 0.2% Tween-20 prepared in H50 buffer which contained 20 mM HEPES (pH 7.5) and 50 mM KCl. 30 μL 0.2 mg/mL neutravidin (dissolved in H50 buffer) was bound to the biotin-BSA. For each flow-through chamber,

non-specific sample binding to the slide was checked in the absence of neutravidin. Ribosomal complexes were imaged in polyamine buffer (50 mM HEPES (pH7.5), 6 mM Mg$^{2+}$, 6 mM β-mercaptoethanol, 150 mM NH$_4$Cl, 0.1 mM spermine and 2 mM spermidine) with 0.8 mg/mL glucose oxidase, 0.625% glucose, 1.5 mM 6-hydroxy-2,5,7,8-tetramethylchromane-2-carboxylic (Trolox) and 0.4 µg/mL catalase. smFRET data were acquired with 100 ms time resolution.

IDL software (ITT) was used to extract flourescence intensities of Cy3 donor ($I_{D/italic}$) and Cy5 acceptor ($I_{A/italic}$), from which apparent FRET efficiency ($E_{FRET/italic}$, hence referred as FRET) was calculated:

$$E_{FRET} = \frac{I_A}{I_A + I_D}$$

Traces showing single-step photobleachings for both Cy5 and Cy3 were selected using MATLAB scripts. FRET distribution histograms compiled from hundreds of smFRET traces were smoothed with a 5-point window using MATLAB and fit to two Gaussians corresponding to 0.4 and 0.6 FRET states (*Ling and Ermolenko, 2015*; *Cornish et al., 2008*; *Ermolenko et al., 2007*). To determine rates of fluctuations between 0.4 and 0.6 FRET states, smFRET traces were idealized by 2-state Hidden Markov model (HMM) using HaMMy software (*McKinney et al., 2006*).

Ribosome complexes used in smFRET experiments were assembled as follows. To fill the P site, 0.3 µM S6/L9-labeled ribosomes were incubated with 0.6 µM tRNA and 0.6 µM mRNA in polyamine buffer at 37°C for 15 min. To bind aminoacyl-tRNA to the ribosomal A site, 0.6 µM aminoacyl-tRNA were pre-incubated with 10 µM EF-Tu and 1 mM GTP in polyamine buffer at 37°C for 10 min. Then, 0.3 µM ribosomal complex containing peptidyl-tRNA in the P site was incubated with 0.6 µM aminoacyl-tRNA (complexed with EF-Tu•GTP) at 37°C for 5 min. For the mixture of all *E. coli* tRNAs (*Figure 5—figure supplement 1*), 30- (0.9 µM) or 150-fold (4.5 µM) molar excess of total aminoacyl-tRNAs (charged with all amino acids except for Tyr) were incubated with 30 nM ribosomes. After the incubation, ribosome samples were diluted to 1 nM with polyamine buffer, loaded on the slide and immobilized by neutravidin and biotinylated DNA oligo annealed to the handle sequence of the ribosome-bound model mRNA. To catalyze translocation, 1 µM EF-G•GTP was added to the imaging buffer.

To prepare dnaX_Slip mRNA-programmed ribosomes that contained N-Ac-Val-Lys-tRNA$^{Lys}$ in the P site (*Figure 2*), N-Ac-Val-tRNA$^{Val}$ and Lys-tRNA$^{Lys}$ were bound to the P and A sites of the S6/L9-labeled ribosome, respectively, as described above. After complex immobilization on the slide and removal of unbound Lys-tRNA$^{Lys}$, ribosomes were incubated with 1 µM EF-G•GTP at room temperature for 10 min. Next, EF-G•GTP was replaced with the imaging buffer and a mixture of 1 µM of EF-Tu•GTP•Lys-tRNALys and 1 µM EF-G•GTP (in imaging buffer) was delivered at 0.4 mL/min speed by a syringe pump (J-Kem Scientific) after 10 s of imaging.

## Puromycin assay

0.6 µM 70S ribosomes were incubated with 1.2 µM dnaX_NS mRNA and 1.2 µM N-Ac-[$^3$H]Phe-tRNA-$^{Phe}$ in polyamine buffer at 37°C for 15 min followed by 10 min incubation with 1 mM puromycin. The puromycin reaction was terminated by diluting the ribosome samples using MgSO$_4$-saturated 0.3 M sodium acetate (pH 5.3), and the N-Ac-[$^3$H]-Phe-puromycin was extracted ethyl acetate.

## Filter-binding assay

The filter-binding assay was performed as previously described (*Salsi et al., 2016*; *Spiegel et al., 2007*) with minor modifications. Ribosome complexes were assembled with radiolabeled tRNAs ([$^{14}$C]Val-tRNA$^{Val}$, [$^3$H]Phe-tRNA$^{Phe}$, [$^3$H]Tyr-tRNA$^{Tyr}$, [$^3$H]Glu-tRNA$^{Glu}$ and N-Ac-[$^3$H]Tyr-tRNA$^{Tyr}$ as indicated in figure legends) similarly to smFRET experiments described above. Ribosome complexes were applied to a nitrocellulose filter (MiliporeSigma), which was subsequently washed with 500 µl (for complexes programmed with dnaX mRNA) or 800 µl (for complexes programmed with HIV mRNA) of ice-cold polyamine buffer containing 20 mM Mg$^{2+}$ to remove unbound tRNAs. 20 mM Mg$^{2+}$ concentration was used to stabilize ribosome complexes under non-equilibrium conditions.

## Frameshifting assay

0.6 µM 70S ribosomes were incubated with 1.2 µM dnaX_Slip mRNA and 1.2 µM *N*-Ac-Val-tRNA$^{Val}$ in polyamine buffer at 37°C for 15 min. The ribosomes were then incubated with 4 µM EF-G•GTP, 10 µM EF-Tu•GTP, 2.4 µM Lys-tRNA$^{Lys}$, 1.2 µM Arg-tRNA$^{Arg}$ (binds in 0 frame) and 1.2 µM [$^3$H]Glu-tRNA$^{Glu}$ (binds in - one frame) at 37°C for 6 min. Incorporation of [$^3$H]Glu into the ribosome was measured by filter-binding assay as described above. Frameshifting efficiency (ribosome A-site occupancy by [$^3$H]Glu-tRNA$^{Glu}$) was normalized by the P-site occupancy of *N*-Ac-[$^3$H]Glu-tRNA$^{Glu}$ non-enzymatically bound to the ribosome programmed with dnaX_Slip ΔFSS mRNA.

## HIV mRNA-70S ribosome complex assembly for cryo-EM analysis

The 70S ribosomes re-associated from 30S and 50S subunits were purified using sucrose gradient. 0.4 µM 70S ribosomes were bound with 0.7 µM *N*-Ac-Phe-tRNA$^{Phe}$ and 0.8 µM HIV_NS (GAG) mRNA in polyamine buffer.

## Cryo-EM and image processing

C-flat grids (Copper, 1.2/1.3, Protochips) were glow-discharged for 30 s in a PELCO glow-discharge unit at 15 mA. 3 µl of the 70S•HIV FSS-mRNA complex at 250 nM concentration were applied to the grid and incubated for 30 s before vitrification using an FEI Vitrobot Mark IV (ThermoFisher). The grids were blotted for 3 s using blotting force 3 at 4°C and ~90% humidity, plunged in liquid ethane, and stored in liquid nitrogen.

A dataset was collected using SerialEM (*Mastronarde, 2005*) on a Titan Krios operating at 300 kV and equipped with a K2 Summit camera (Gatan). A total of 5208 movies were collected using three shots per hole in super-resolution mode and a defocus range of −0.5 to −2.5 µm. The exposure length was 75 frames per movie yielding a total dose of 75 e-/ Å$^2$. The super-resolution pixel size at the specimen level was 0.5115 Å. All movies were saved dark-corrected.

Gain and dark references were calculated using the method described by *Afanasyev et al., 2015* and used to correct the collected movies in cisTEM (*Grant et al., 2018*). All further image processing was done using cisTEM. The movies were magnification-distortion-corrected using a calibrated distortion angle of 42.3° and a scale factor of 1.022 along the major axis and binned by a factor of 2. The movies were motion-corrected using all frames, and CTF parameters were estimated. Particles were picked using the particle picker tool in cisTEM and then curated manually. A total of 640,261 particles were extracted in 648$^2$ pixel boxes.

Extracted particles were aligned to an unpublished reference volume using a global search in the resolution range from 8 to 300 Å (for classification workflow, see *Figure 8—figure supplement 1*). The resulting 3D reconstruction was calculated using 50% of the particles with the highest scores and had a resolution of 3.27 Å (Fourier Shell Correlation = 0.143). Next, classification into eight classes without alignment with a focus mask around the A-, and P-sites of the large and small subunit yielded two classes with density in the A-site. The classes corresponded to one rotated (23.15% of all particles), and one non-rotated state (11.44% of all particles), respectively. Both states were extracted separately and refined using local refinement with increasing resolution limits to 5 Å followed by one round of CTF refinement without alignment. The rotated and the non-rotated states reached resolutions of 3.15 Å and 3.35 Å, respectively. Each class then was classified into five classes without alignment using a focus mask around the observed density in the A-site. Two classes obtained from the non-rotated state showed weak density in the A-site. The two classes were merged and classified further into eight classes. Two classes had A-site density of which one showed strong density corresponding to the hairpin in the A-site and tRNA$^{Phe}$ in the P-site. Particles for this class were extracted and aligned with increasing resolution limits to 5 Å. Finally, CTF refinement to 4 Å resolution without alignment and a step size of 50 Å was run and the final reconstructions were calculated using a beam-tilt corrected particle stack yielding final resolutions of 3.4 Å and 3.3 Å (Fourier Shell Correlation = 0.143).

The classification for the R conformation was done as described for the NR conformation. Classification into five classes yielded two classes with hairpin density. The classes were merged and classified into 8 classes of which four classes had weak density and one class yielded strong density. This class was extracted, CTF, and beam-tilt refined yielding a final resolution of 3.1 Å.

Finally, the obtained maps were sharpened in cisTEM and using the local resolution dependent function in phenix.autosharpen (*Terwilliger et al., 2018*).

## Model building and refinement

As the starting model for refinement we used the structure of the *E. coli* 70S ribosome with a ternary complex (PDB ID 5UYL), omitting EF-Tu and the A-site tRNA. An NMR structure of the HIV-1 frame-shifting element (PDB ID 1PJY) was used as the starting model for the hairpin and to generate secondary-structure restraints. Missing parts of the mRNA were built manually and the geometry was regularized in phenix.geometry_minimization before refinement. The A-site finger was modeled using nucleotides 873–904 from PDB ID 5KPS where the A-site finger is well-ordered. Protein secondary structure restraints were generated in Phenix (*Adams et al., 2010*) and edited manually. We generated base-pairing (hydrogen bonds) restraints using the 'PDB to 3D Restraints' web-server (http://rna.ucsc.edu/pdbrestraints/, [*Laurberg et al., 2008*]) and added stacking restraints manually for the hairpin, and A-site finger.

Initially, the ribosomal subunits, tRNA and the hairpin were separately fitted into the cryo-EM, using Chimera, followed by manual adjustments in Coot (version 0.9-pre) (*Emsley et al., 2010*). The structural model was refined using phenix.real_space_refine (*Afonine et al., 2018*) and alternated with manual adjustments in Coot. The final model was evaluated in MolProbity (*Williams et al., 2018*).

## Acknowledgements

These studies were supported by HHMI (to NG) and grants from the US National Institute of Health R01GM099719 (to DNE), a CFAR pilot award (to DNE) from P30 AI078498 (PI Stephen Dewhurst) and 5R35GM127094 (to AAK). We thank Enea Salsi, Elie Farah and Joshua Mix for their early contributions to this project.

## Additional information

### Competing interests

Nikolaus Grigorieff: Reviewing editor, *eLife*. The other authors declare that no competing interests exist.

### Funding

| Funder | Grant reference number | Author |
| --- | --- | --- |
| National Institute of General Medical Sciences | R01GM099719 | Dmitri N Ermolenko |
| National Institute of General Medical Sciences | 5R35GM127094 | Andrei A Korostelev |
| Howard Hughes Medical Institute | | Nikolaus Grigorieff |
| National Institute of Allergy and Infectious Diseases | P30 AI078498 | Dmitri N Ermolenko |

The funders had no role in study design, data collection and interpretation, or the decision to submit the work for publication.

### Author contributions

Chen Bao, Conceptualization, Data curation, Formal analysis, Investigation, Writing - original draft, Writing - review and editing; Sarah Loerch, Conceptualization, Data curation, Validation, Investigation, Visualization, Writing - original draft, Writing - review and editing; Clarence Ling, Data curation; Andrei A Korostelev, Formal analysis, Funding acquisition, Investigation, Visualization, Writing - original draft, Writing - review and editing; Nikolaus Grigorieff, Dmitri N Ermolenko, Conceptualization,

Funding acquisition, Investigation, Writing - original draft, Project administration, Writing - review and editing

### Author ORCIDs

Chen Bao http://orcid.org/0000-0002-9224-8083
Sarah Loerch http://orcid.org/0000-0002-1731-516X
Andrei A Korostelev http://orcid.org/0000-0003-1588-717X
Nikolaus Grigorieff https://orcid.org/0000-0002-1506-909X
Dmitri N Ermolenko https://orcid.org/0000-0002-7554-5967

### Decision letter and Author response

Decision letter https://doi.org/10.7554/eLife.55799.sa1
Author response https://doi.org/10.7554/eLife.55799.sa2

## Additional files

### Supplementary files

• Supplementary file 1. Model mRNA sequences. The SD sequence is shown in green, slippery sequence in magenta, FSS in red, and handle sequence, which is complementary to biotinylated DNA oligo, underlined. To change the linker length between the HIV FSS and the P-site codon, HIV_NS$_{12-nt\ linker}$ mRNA and HIV_NS$_{13-nt\ linker}$ mRNA were prepared by extending the linker from native 11 nucleotides to 12 and 13 nucleotides, respectively. To alter codon identities, a UAC-to-GAG mutation was made on HIV_NS mRNA to generate HIV_NS (GAG) mRNA. Similarly, AUG-to-GAG and UUC-to-AUG mutations were made to generate HIV_NS (AUG) mRNA. The codon replacements are colored orange. Vertical black bars indicate the 3' ends of ΔFSS mRNAs.

• Transparent reporting form

### Data availability

Structural models have been deposited in PDB under the accession codes 6VWM, 6VWN, 6VWL. Cryo-EM data have been deposited to EMDB under the accession codes EMD-21421, EMD-21422, EMD-21420.

The following datasets were generated:

| Author(s) | Year | Dataset title | Dataset URL | Database and Identifier |
|---|---|---|---|---|
| Loerch S, Bao C, Ling C, Korostelev AA, Grigorieff N, Ermolenko DN | 2020 | 70S ribosome bound to HIV frameshifting stem-loop (FSS) and P-site tRNA (nonrotated conformation, Structure I) | http://www.rcsb.org/structure/6VWM | RCSB Protein Data Bank, 6VWM |
| Loerch S, Bao C, Ling C, Korostelev AA, Grigorieff N, Ermolenko DN | 2020 | 70S ribosome bound to HIV frameshifting stem-loop (FSS) and P-site tRNA (nonrotated conformation, Structure I) | http://www.ebi.ac.uk/pdbe/entry/emdb/EMD-21421 | Electron Microscopy Data Bank, EMD-21421 |
| Loerch S, Bao C, Ling C, Korostelev AA, Grigorieff N, Ermolenko DN | 2020 | 70S ribosome bound to HIV frameshifting stem-loop (FSS) and P-site tRNA (nonrotated conformation, Structure II) | http://www.rcsb.org/structure/6VWN | RCSB Protein Data Bank, 6VWN |
| Loerch S, Bao C, Ling C, Korostelev AA, Grigorieff N, Ermolenko DN | 2020 | 70S ribosome bound to HIV frameshifting stem-loop (FSS) and P-site tRNA (nonrotated conformation, Structure II) | http://www.ebi.ac.uk/pdbe/entry/emdb/EMD-21422 | Electron Microscopy Data Bank, EMD-21422 |
| Loerch S, Bao C, Ling C, Korostelev AA, Grigorieff N, Ermolenko DN | 2020 | 70S ribosome bound to HIV frameshifting stem-loop (FSS) and P/E tRNA (rotated conformation) | http://www.rcsb.org/structure/6VWL | RCSB Protein Data Bank, 6VWL |
| Loerch S, Bao C, Ling C, Korostelev AA, Grigorieff N, | 2020 | 70S ribosome bound to HIV frameshifting stem-loop (FSS) and P/E tRNA (rotated conformation) | http://www.ebi.ac.uk/pdbe/entry/emdb/EMD-21420 | Electron Microscopy Data Bank, EMD-21420 |

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
