## [Decision Letter]

Thank you for submitting your article "mRNA stem-loops can pause the ribosome by hindering A-site tRNA binding" for consideration by *eLife*. Your article has been reviewed by two peer reviewers, one of whom is a member of our Board of Reviewing Editors, and the evaluation has been overseen by James Manley as the Senior Editor The reviewers have opted to remain anonymous.

The reviewers have discussed the reviews with one another and the Reviewing Editor has drafted this decision to help you prepare a revised submission.

Summary:

This paper probes the molecular mechanism of structured frameshifting stimulatory sequences (FSSs) in stimulating -1 programmed ribosome frameshifting (PRF) on the bacterial dnaX slippery site. Single-molecule FRET assays that monitor the intersubunit rotation of the large and small subunits to determine the rate of EFTu-GTP-tRNA TC binding to the A site and then EF-G stimulated translocation show that positioning of the FSS in the mRNA entry channel slows down both reactions. The effect on translocation rates had been seen before, but the slowing of A-site binding of the TC in the zero frame appears to be a novel finding. This effect was dependent on positioning of the FSS at the entry channel. The same was demonstrated for the FSS of the HIV -1 PRF site that could be reconstituted in vitro with *E. coli* components, and was shown to occur with a different A-site codon/TC pair. Filter binding assays of TC loading were used to support the conclusion that FSSs inhibit A-site tRNA binding only when positioned near the opening of the mRNA entry channel. They go on to present a cryoEM structure of a 70S ribosome bound to a variant of the HIV PRF sequence in which the slippery codons were replaced by UUC-GAG codons with Phe-tRNA in the P site. Interestingly, the FSS occupies the A-site, providing a structural explanation for its ability to inhibit TC binding to the A-site. This structure differs from one presented recently for the dnaX PRF sequence in which the FSS was found just outside the entry channel opening. They propose a model in which either of these locations of the FSS sequence can interfere with A-site binding, and the empty A site would allow frameshifting through one of two proposed mechanisms for frameshifting involving slippage of the single P-site tRNA when the A-site is vacant.

Essential revisions:

It is necessary to revise the interpretation of the results and include the appropriate alternative possibilities and the qualifications described in reviewer #2's major comments. Owing to the COVID-19 pandemic, it is not required that the control experiments requested by reviewer #1 be conducted, but additional text relating to the missing data should be included.

Reviewer #1:

The work is well executed and interdisciplinary, and generally well described, and the results provide significant insights into the molecular mechanism of frameshift stimulation by FSS elements.

The authors need to include the binding of labeled Phe-tRNA or Val-tRNA as controls for the experiments in Figure 6—figure supplement 1B.

Reviewer #2:

In this manuscript, the authors attack a longstanding and interesting problem in the field of translational control, programmed -1 ribosomal frameshifting. The technical basis for this work involves an in vitro *E. coli* translation system using -1 PRF signals derived from a prokaryotic (dnaX) and eukaryotic (HIV-1) sources. Note that these -1 PRF signals are atypical: they are representative of only a few known -1 PRF signals in which the Frameshift Stimulating Signal (FSS) is a stem loop as opposed to an RNA pseudoknot. This may or may not be germane. Three orthogonal approaches are employed: smFRET, classical tRNA binding biochemistry, and cryo-EM. The methodologies are very strong and cutting edge, and the data all convincingly point to slippage by a single peptidyl-tRNA while the A-site is empty and when the ribosome is in an unrotated state. The data also indicate that this is because the FSS occludes aa-tRNA binding to the A-site.

The data presented here also indicate a non-simultaneous slippage mechanism, i.e. a 1 tRNA slip event. While it has been shown that such slips can happen (using the HIV-1 PRF signal in *E. coli*, see work from the Brakier-Gingras group), this is "non-canonical. Indeed, the more commonly accepted "simultaneous slippage model" proposed by the Varumus lab in 1988 posited that slippage occurs with a ribosome in which both the 0-frame A- and P-sites slip over the slippery site. However, as described below, that paper did indicate that ~30% of -1 PRF was due to single tRNA slippage.

Perplexingly, this in contrast to other recent work that suggests that slippage happens a hyper-rotated state during which a ribosome with tRNAs in both the A-and P-sites are occupied by tRNAs corresponding to the 0-frame slippery site, and during which translocation of the A- and P-site tRNAs at the slippery site. For example, the Cornish and Pugilsi labs have shown that frameshifted ribosomes have extended stays in the rotated (or even hyper-rotated) state. Here, the data indicate that ribosomes are in the non-rotated state. Is the difference because frameshifting was decoupled from FSS induced pausing?

Qin et al., 2014; Chen et al., 2014. Additionally, the Rodnina lab's recent work supports this view as well.

This not irresolvable. Earlier studies from many labs support the hypothesis that there are at least three different kinetic pathways to frameshifting: 1) slippage of the peptidyl-tRNA with an empty A-site, 2) simultaneous slippage of two tRNAs during accommodation, and 3) simultaneous slippage as they translocate out of the slippery site.

It is possible that different experimental designs favor one pathway over another. This one for example seems to favor pathway 1. Mutants of eEF1A (and presumably of EF-Tu) illuminate pathway 2. And the setups used by the Rodnina, Giedroc and Puglisi labs seem to favor number 3.

Indeed, if we look at Jacks and Varmus 1988 paper, they show evidence supporting all three pathways (see Figure 2 of that paper). Their mass spec analysis of the frameshift fusion peptide show that 70% of the products include the 2 zero-frame amino acids (supportive of pathways 2 and/or 3), and that 30% have the zero-frame P site amino acid and the -1 frame A-site encoded amino acid (supporting pathway 2). Work from the Brakier-Gingras lab (the HIV-1 frameshift signal assayed in *E. coli*) strongly favored pathway 1. Work from the Dinman lab, using frameshift signals in yeast, presented data supporting all three. That group also did an interesting set of simulations describing how each of the three pathways can be favored by changing various kinetic parameters.

It is possible that this particular experimental setup, where the frameshift signal is extremely close to the start codon, may be influencing the preferred pathway. Earlier genetic studies showed that putting frameshift signals close to start codons inhibited -1 PRF. More recent work from the Djuranovic lab has identified a short translational ramp after initiation that affects protein synthesis. It may also be an artifact of the translational system that is used: *E. coli* ribosomes are smaller than eukaryotic ribosomes. Maybe their different footprints and/or differences in the mRNA entry tunnel (e.g. differences in RNA helicase activities?) may be swing the balance to favor one pathway over another.

Lastly, one has to question the assumption that substitution of non-slippery codons for slippery ones in the "slippery site" does not affect the molecular mechanism underlying -1 PRF. I think that this should not be taken as a matter of faith.

In sum, I think that this manuscript is technically robust and technologically state-of-the-art. They have certainly illuminated some aspect of -1 PRF, but it has certainly not solved all of the questions. Indeed, it raises many more (which is what good science does). My concern is that the authors knowledge of the -1 PRF literature is not as deep as it could be, and that as a result their interpretation of the data is not properly contextualized.

---

## [Author Response]

Reviewer #1:The work is well executed and interdisciplinary, and generally well described, and the results provide significant insights into the molecular mechanism of frameshift stimulation by FSS elements.The authors need to include the binding of labeled Phe-tRNA or Val-tRNA as controls for the experiments in Figure 6—figure supplement 1B.

smFRET and filter-binding experiments (Figure 2-4, Figure 6A-B of revised manuscript) indicate that EF-Tu-dependent binding of Phe-tRNA^Phe^ to the A site is not affected when the dnaX FSS is 15 nucleotides downstream of the P site. Since our labs are shut down due to the COVID-19 crisis, we cannot do the requested additional filter-binding measurements of non-enzymatic binding of N-acetyl-Phe-tRNA^Phe^.

Reviewer #2:The data presented here also indicate a non-simultaneous slippage mechanism, i.e. a 1 tRNA slip event. While it has been shown that such slips can happen (using the HIV-1 PRF signal in *E. coli*, see work from the Brakier-Gingras group), this is "non-canonical. Indeed, the more commonly accepted "simultaneous slippage model" proposed by the Varumus lab in 1988 posited that slippage occurs with a ribosome in which both the 0-frame A- and P-sites slip over the slippery site. However, as described below, that paper did indicate that ~30% of -1 PRF was due to single tRNA slippage.Perplexingly, this in contrast to other recent work that suggests that slippage happens a hyper-rotated state during which a ribosome with tRNAs in both the A-and P-sites are occupied by tRNAs corresponding to the 0-frame slippery site, and during which translocation of the A- and P-site tRNAs at the slippery site. For example, the Cornish and Pugilsi labs have shown that frameshifted ribosomes have extended stays in the rotated (or even hyper-rotated) state. Here, the data indicate that ribosomes are in the non-rotated state. Is the difference because frameshifting was decoupled from FSS induced pausing?[…]In sum, I think that this manuscript is technically robust and technologically state-of-the-art. They have certainly illuminated some aspect of -1 PRF, but it has certainly not solved all of the questions. Indeed, it raises many more (which is what good science does). My concern is that the authors knowledge of the -1 PRF literature is not as deep as it could be, and that as a result their interpretation of the data is not properly contextualized.

We thank the reviewer for these insightful comments and for appreciating that our work is technically robust.

Presence of multiple pathways of -1PRF in our data. We agree that the three mechanisms of -1PRF have been considered in the field: (i) slippage of the single P-site tRNA when the A site remains vacant, (ii) frameshifting of both A-site and P-site tRNAs during aa-tRNA accommodation to the A site, and (iii) slippage of A- and P-site tRNAs during translocation (Dinman, 2012). Multiple lines of evidence indicate that these pathways could be active at the same time. Our data also support this idea.

Our smFRET experiments performed with dnaX_Slip mRNA showed FSS-induced ribosome pausing in both non-rotated and rotated states (Figure 2), indicating that the FSS inhibits both A-site tRNA binding and translocation. The ribosome pausing in the rotated state is consistent with the work of the Puglisi and Rodnina labs. As suggested by the reviewer and discussed in our manuscript, our observation that FSSs induce ribosome pausing in both non-rotated and rotated states is consistent with single tRNA slippage (i) and translocation (iii) frameshifting pathways, respectively. Because pausing in the non-rotated state caused by inhibition of tRNA binding is a novel observation, we focused the rest of our study on this.

In the original manuscript, we tried to streamline our discussion of -1 PRF mechanisms because our work is mainly focused on one aspect of -1 PRF – mechanisms by which frameshift-inducing stem-loops slow down the ribosome. Although we cited many of the works mentioned by the reviewer in the previous version of manuscript, we agree that discussion of -1 PRF mechanisms in the manuscript should be expanded. We added a paragraph to Discussion to address the reviewer’s concerns:

“Our findings provide new insights into the mechanisms of -1PRF stimulated by stem-loop FSSs. […] It is not clear whether pseudoknot FSSs also utilize both pathways or A-site inhibition/P-site tRNA slippage is unique to stem-loop FSSs”.

Experimental conditions may favor one -1PRF pathway over the other. We find that FSSs of both bacterial (dnaX) and eukaryotic virus (HIV) origins inhibit A-site binding in *E. coli* ribosomes. Hence, RNA hairpin competition with tRNA binding is unlikely an artifact of using *E. coli* ribosomes with the stem-loop from eukaryotic virus. Nevertheless, we agree with the reviewer that depending on experimental conditions (primarily tRNA abundance as indicated by works of Rondina’s group), one frameshifting pathway may be favored over the other.

Ribosome pausing in the presence of non-slippery codons. We observe that the dnaX FSS inhibits A-site tRNA binding in the context of both slippery and non-slippery sequences. Hence, our discovery of mRNA stem-loop binding to the A site may have broader implications for translation regulation not limited to -1 PRF.